# GIVE: Structured Reasoning with Knowledge Graph Inspired Veracity Extrapolation

## Abstract

Existing retrieval-based frameworks to enhance large language models (LLMs) requires accessibility to a rich non-parametric knowledge source which contains factual information that directly solves the query. Reasoning based approaches heavily rely on the parametric knowledge of the model to provide domain-specific explicit reasoning chain. However, inclusive knowledge sources are expensive or infeasible to build for scientific or corner domains, thus not applicable in either training or inference time. To tackle the challenges, we introduce Graph Inspired Veracity Extrapolation (GIVE), a novel reasoning framework that integrates the parametric and non-parametric memories to enhance both knowledge retrieval and faithful reasoning processes using very limited external clues. By leveraging the structured knowledge to inspire LLM to model the interconnections among relevant concepts, our method facilitates a more logical and step-wise reasoning approach akin to experts' problem-solving, rather than gold answer retrieval. Specifically, the framework prompts LLMs to decompose the query into crucial concepts and attributes, construct entity groups with relevant entities, and build an augmented reasoning chain by probing potential relationships among node pairs across these entity groups. Our method incorporates both factual and extrapolated linkages to enable comprehensive understanding and response generation. Extensive experiments on domain-specific and open-domain benchmarks demonstrate the effectiveness of our proposed method, thereby underscoring the efficacy of integrating structured information and internal reasoning ability of LLMs for tackling difficult tasks with limited external resources.

## 1 INTRODUCTION

Large language models (LLMs) (Ouyang et al., 2022; et al., 2024a;b; 2022; 2020; Raffel et al., 2020) have been shown to be able to generate fluent language, answer questions and induce knowledge from the given text in recent benchmarks. Though it shows a great performance for general question answering, we do not see a similar level of success on similar tasks under the scientific domains or settings that require specialized knowledge tailored to a certain context (Cai et al., 2024; Zhang et al., 2024; Dorfner et al., 2024; Dong et al., 2023; Zhong et al., 2023; Deng et al., 2024). Two technical disadvantages of LLMs might explain the unsatisfactory performance. On the one hand, LLMs are not aware of specialized domain knowledge (Ge et al., 2023; Bang et al., 2023; Xie et al., 2024), such as protein-gene relations, drug-disease associations, and actor profiles of newly introduced movies. The specialized knowledge is not obtained through training and needs to be constantly updated. On the other hand, LLMs are not equipped to lay out a multi-step logic chain with domain expertise to identify and solve sub-questions following a correct thinking process (Wei et al., 2023; Wang et al., 2023; Jiang et al., 2024). For example, to identify whether a drug is capable of treating a disease, the model needs to figure out the root cause and the affecting organ of the disease, chemicals that interact with the virus, and the drug formulas that contain the correct chemicals. These thinking processes are unique to questions and are hardly presented in instruction tuning or human preference alignment training data.

Existing works use LLM to guide the factual knowledge search on well-populated external knowledge structures like knowledge graphs (KG) (Sun et al., 2024; Xu et al., 2024). However, expert-curated domain-specific knowledge graphs such as UMLS are expensive to obtain and update. Enhancing LLM reasoning with a sparse external KG is a more realistic and practical setting. Though the

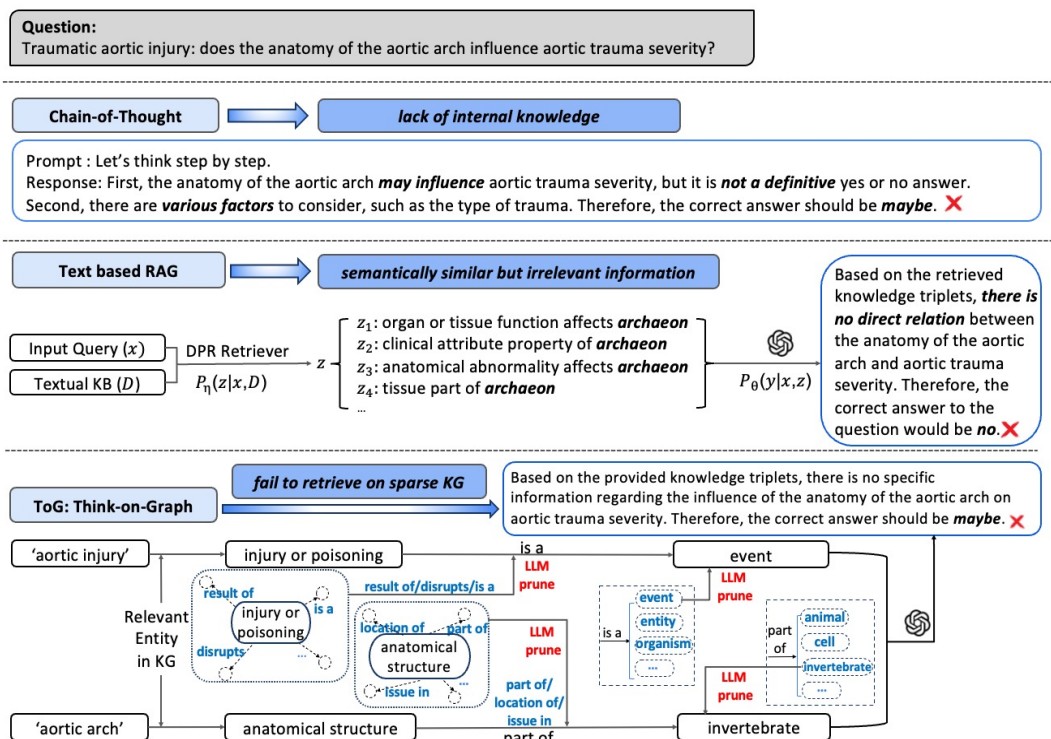

Figure 1: An example from PubmedQA. Without gold context, Chain-of-Thought (CoT) fails because LLM's internal knowledge fails to form a faithful logic chain. Retrieval Augmented Generation (RAG) retrieves semantically similar but irrelevant information on a sparse KG, leading to hallucination.Think on Graph (ToG) focuses on using internal knowledge of LLM to prune the external information, thus fails on sparse KG for lack of high-quality candidates.

incomplete KGs lack evidence that contributes to problem-solving directly, they encapsulate the intuition and experience of the curation experts during knowledge structure construction, such as feasible relation collection and possible connections among similar entities.

In this work, we aim to address the limitations of reasoning exclusively on internal knowledge aor external knowledge frameworks by proposing GIVE, a graph-inspired veracity extrapolation framework. GIVE simulates the thinking processes of the KG constructors and utilizes the structure of KG as inspiration. It populates the sparse KG with silver edges by receiving hints from factual connections, concretizing the internal knowledge of LLMs, constructing counterfactual reasoning to combat hallucinations, and additionally retrieving existing related evidence on KG if needed. GIVE first obtains a focused set of entities that are mostly related to the question by prompting LLMs. Using the potential relations between the relevant KG concepts, we construct a reasoning framework including all possible concepts and their potential connections that could facilitate question answering. We introduce additional intermediate node groups by picking the multi-step reasoning plans that are most helpful for the ultimate questions. GIVE includes factual connections backed by KG, internal knowledge obtained through pre-training, and novel relations that bridge similar concepts from the veracity extrapolation process. To complete the reasoning framework, we also incorporate counterfactual connections among nodes to prevent hallucination. Ultimately, we develop a method that (1) retrieves external knowledge for more informed question answering; (2) induces a structured reasoning processes by extrapolating KG triplets to related queried concepts, which we refer to as "veracity extrapolation".

We experiment with our proposed method on biomedical, commonsense, and truthfulness question answering. GIVE uniformly achieves the best performance on QA tasks of different domains and types, utilizing KGs of different sizes and sparcities, among all internal knowledge/external knowledge reasoning baselines, indicating the effectiveness and robustness of the proposed framework. GIVE pioneers in boosting LLM's reasoning ability using very limited external clues to excite its own problem solving ability.

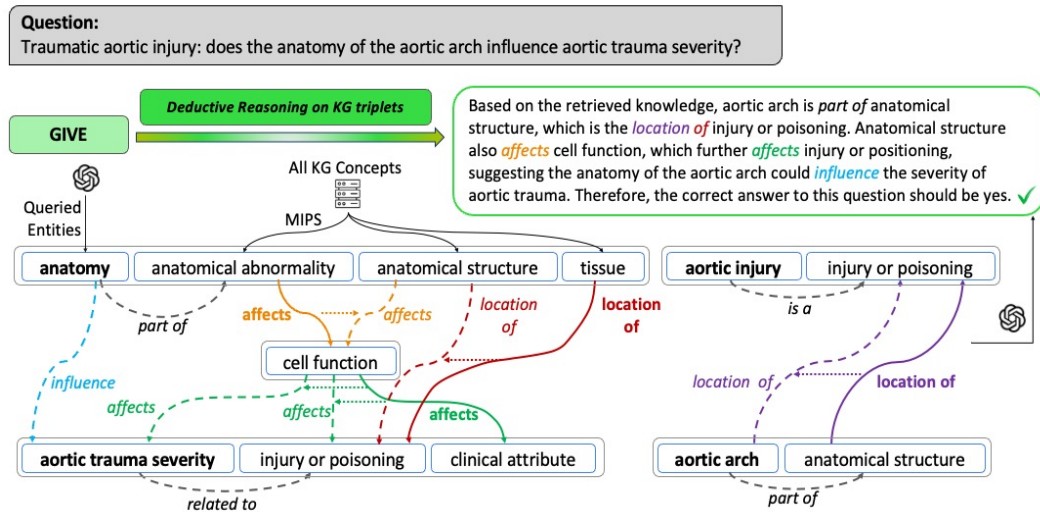

Figure 2: Reasoning process of the proposed method GIVE. Solid lines are the expert KG information, dashed lines are the result of our "veracity extrapolation" process. ; GIVE first builds an entity group for each queried concepts, then induce inner-group connections using its internal knowledge. The cross-group connections contained in the KG are treated as evidence to guide LLM to extrapolate the veracity of such possible relationships between other similar cross-group concepts. A deep and faithful logic chain ending at the queried entities is thus formed by bridging these inner-group and cross-group connections. "cell function" in this case is considered as intermediate entities to facility multi-step reasoning.

## 2 PRELIMINARIES

**Reasoning on Structured Knowledge Base.** Structured knowledge bases like knowledge graphs (KG) provides better knowledge traceability and correct-ability due to the structured nature of the knowledge source, thus provides the RAG-framework with better flexibility during knowledge retrieval. Previous studies encode the KGs (Saxena et al., 2020; Zan et al., 2022) and queries, answer is generated using similarities between node embedding and query embedding. To contrast, we propose a one-shot solution for information retrieval from sparse KG and structured reasoning chain development without any training cost. Recently, ToG (Sun et al., 2024) proposes to iteratively query LLM to search and prune the optimal knowledge paths to include; GoG (Xu et al., 2024) decomposes the query into a set of sub-questions and prompts LLM to iteratively solve each of them. GNN-RAG (Mavromatis & Karypis, 2024) formulates the answer-extraction process as a node classification problem over the knowledge graph. These methods, although proved to be effective in answering queries in specific KG-QA pairs, are built on the assumption that the high-quality data base that contains the gold knowledge are always accessible and easy to retrieve. In the context of scientific domains, however, building specific knowledge bases is challenging, because it requires advances in both domain-specific natural language processing(NLP) and filed-wide vocabulary standardization (Badal et al., 2019; Verhagen et al., 2012). General knowledge bases like Wikipedia or Freebase contains tens of millions of irrelevant entities and triplets, thus are time and resource consuming for LLMs to search from.

**Problem Definition.** In this paper, we study reasoning-rich domain-specific question answering using very sparse knowledge graphs. A knowledge graph (KG) is defined as $G = \{E_G, R_G, \mathcal{E}_G\}$, $\mathcal{E}_G = \{(u, r, v), u, v \in E_G, r \in R_G\}$, where $E_G$ is the set of entities and $R_G$ is the set of relations. An input query $x$ is a statement about the entities $E_x$ and relations $R_x$. For example, the query "Is melatonian effective for insomnia?" contains entity set $E_x = \{$melatonian, insomnia$\}$ and relation set $R_x = \{$effective for$\}$.

To solve an instance of $(G, x)$, the key step is to retrieve highly-relevant knowledge set from the KG. Suppose the gold knowledge set $\mathcal{T}_G(x)$ for x in $G$ is the collection of knowledge triplets in G that explicitly contains the ground-truth output for x. Previous works focus on the case when the given knowledge graph $G$ contains some gold knowledge about x, i.e. $G \in \mathcal{G}_x^*$. We provide a solution to the general form of KGQA instances (G,x) where $G \notin \mathcal{G}_x^*$, by first combining the parametric memory of LLM and the sparse KG $G$, query $x$ to expand the entity and relation set to consider, before using the structure of G to inspire LLM to extrapolate the veracity of the generalised edges among all related

Figure 3: A detailed example of the proposed Veracity Extrapolation process: The gold KG $G^*$ contains the gold knowledge set $\mathcal{T}_{G^*}(x)$, but is infeasible to build. Directly retrieving knowledge $\mathcal{T}_G(x)$ from the accessible KG G results in hallucination. GIVE tackles this challenge by building the augmented entity groups combining KG concepts and queried entity, probing potential relations across the related concept pairs based on the queried relation and KG relation, then use LLM to prune the valid factual and counter-factual candidate knowledge, thus prompt LLM to generate faithful CoT.

concepts, thus solve the input query. Formally, our approach is formulated as:

$$p(y|x,G) \coloneqq p_\alpha(\mathbb{N}_x, \mathbb{R}_x | x, G) p_\beta(\tilde{\mathcal{T}}_x(G)|x, \mathbb{N}_x, \mathbb{R}_x) p_\gamma(y|x, \tilde{\mathcal{T}}_x(G)) \tag{1}$$

where $\mathbb{N}$ and $\mathbb{R}$ are the expanded sets of relevant entities and relations, $\tilde{\mathcal{T}}_x(G)$ is an extrapolated knowledge set that combines the external evidence in $G$ and $x$, and the parametric internal memory of LLM.

## 3 GIVE: GRAPH-INSPIRED VERACITY EXTRAPOLATION

Our proposed method prompts faithfully inductive reasoning by 1) Using LLM to decompose the query into important concepts and attributes; 2) constructing entity groups by combining the key entity in the query and its relevant concepts in the knowledge graph; 3) inducing inner-group connections between the queried entity and related concepts using parametric knowledge of LLM; 4) build inter-group connections by probing and pruning all pairwise possible connections, and considering intermediate concept groups to facilitate multi-hop reasoning for complicated questions. We present the overall algorithm for GIVE in Appendix 1 and the prompt and example output in Appendix G.

### 3.1 QUERY INFORMATION EXTRACTION

Given query x, GIVE first leverages the LLM to retrieve the entity and relation sets $E_x$ and $R_x$:

$$x \to LLM \to E_x, R_x \tag{2}$$

where $E_x = \{e_x^0, e_x^1 ... e_x^n\}$ denotes the top-k concepts, and $R_x = \{r_x^0, r_x^1 ... r_x^m\}$ is the top-m relations or attributes in the query.

### 3.2 ENTITY GROUP CONSTRUCTION

The goal of this step is to bridge the gap between the limited richness of knowledge base corpus and the complexity of the potential input. To this end, we search through the knowledge space, to construct a cluster of similar concepts, for each of the entities that we identified as important for the given query. For each $e_x^k \in E_x$, GIVE leverages an underlying pre-trained LM encoder $w$ to encode the concepts in the knowledge base, and retrieve p most similar concepts to each queried entity by comparing cosine similarities:

$$Y_x^k = \{y_{x\,1}^k, y_{x\,1}^k ... y_{x\,p}^k\} = \operatorname*{argmin}_{\hat{y} \in E_G} {}_p\{cos(w(e_x^k), w(\hat{y}))\} \tag{3}$$

The set $Y_x^k$ contains all entities that are semantically similar to the queried concept $e_x^k$, and $e_x^k$ is appended to $Y_x^k$ to formulate the entity group $N_x^k$:

$$N_x^k = \{e_x^k\} \cup Y_x^k \quad \mathbb{N}_x = \{N_x^i\}_{i=1}^k \tag{4}$$

There are two advantages of the entity set $N_x^k$s: (1) Inducing inter-group connections between the queried entity and its "sibling" concepts naturally lead to a reasoning chain on the KG concepts. (2) It relaxes the strict information retrieval on the few queried entity, to relationship inference over a larger set of relevant concepts.

## 3.3 INNER-GROUP CONNECTIONS

Firstly, considering each $N_x^i$ in Section 3.2, we induce connections between each queried entity and its other semantically-similar concepts in its own entity group. The purpose of this step is to inspire LLM to conduct divergent thinking on the related similar concepts, not only focus on the queried entity itself. The hard problem of inducing relationships directly between two queried entities is released to finding any possible relations between two sets of similar concepts.

To this end, we utilize LLM to openly fill in the relationship between the queried entity and each of the in-group concept. Consider an entity group that consists of 1 queried entity and $p$ additional KG concepts $N_x^k = \{e_x^k, y_{x1}^k, y_{x2}^k...y_{xp}^k\}$, for $1 \le m \le p$:

$$(e_x^k, y_{xm}^k) \to LLM \to (e_x^k, r_m^k, y_{xm}^k) \tag{5}$$

## 3.4 INTER-GROUP CONNECTIONS

In this section, we provide evidence for the language model to induce relationships between the node pairs that cross two entity groups. Given two concept groups $N_x^i$ and $N_x^j$, we first identify all legitimate relations that could be used to connect any node pairs across these two groups, and use LLM to prune these psedu-connections. We also consider intermediate groups to facilitate multi-hop reasoning.

### 3.4.1 POTENTIAL RELATIONS INDUCTION

Between each pair of entity groups, we consider two kinds of potential relations: (1)**Relations mentioned in the question.** The relations asked by the ultimate QA task are the critical connections to be considered. We induce the relation in questions by prompting the LLM while providing instruction and examples on identifying relations and the content of the question as input. The relation would be a sub-sequence of the original question. (2)**KG relations that exists between these two groups.** Since each group contains semantically similar concepts, the existing cross-group KG connections could potentially connect nodes in two node groups with correct semantic meaning.

GIVE boosts the reasoning ability of LLM by inspiring it to consider these two kinds of potential relations, when inducing useful knowledge between two entities. Formally, the potential relations $R_x^{ij}$ that could be used to connect the queried entities $e_x^i$ and $e_x^j$ are the queried relations and the relations connects their relevant entities in the knowledge graph:

$$R_G^{ij} = \{r, (u, r, v) \in \mathcal{E}_G, u \in N_i, v \in N_j\} \tag{6}$$

$$R_x^{ij} = R_x \cup R_G^{ij} \quad \mathbb{R}_x = \{R_x^{ij}\}_{i,j=1}^k \tag{7}$$

For example, considering two node groups about "chemical" and "gene", certain chemicals might "upregulate" certain genes and another set of chemicals might be "substrate" for some genes. Combining any relations from the "chemical" group to the "gene" group, we can identify all feasible relations that could correctly describe the relation between two kinds of nodes according to the knowledge graph would be a set of relations ("upregulate", "downregulate", "agonize", "antagonize" and "serve as substrate"). The combined set of potential relations among each pair of entity groups would facilitate the following process of building connections among nodes.

### 3.4.2 INTERMEDIATE NODE GROUP DISCOVERY FOR MULTI-STEP REASONING

Considering only nodes and connections directly related to the ones mentioned in the question would limit the thinking scope, this is especially the case when dealing with scientific questions when neither LLM nor the external knowledge source itself has enough information to connect two entity groups directly. For example, when answering a query about the effect of certain drug to a disease,

a natural reasoning chain is to build the (drug, compound, disease) connections, to form the claim "because certain compound entity is contained in the drug, and the diseases that can be treated by the entity, it could be inferred that the drug can treat entity".

To provide sufficient knowledge and thinking hints for such complicated tasks that require linking target entities through multi-step reasoning, GIVE explores new node groups as intermediate stopovers of the thinking process. Firstly, all length-2 paths between two node groups are discovered. Secondly, LLM is prompted to automatically select the most helpful multi-hop thinking process that benefits the ultimate question-answering task, where each multi-hop thinking process comes from verbalization of the length-2 path we discovered. Using the intermediate node of the optimal length-2 path, GIVE constructs an intermediate entity group by leveraging the same process as illustrated in Section 3.2. Note that the intermediate node group is created to build the multi-hop connections between two queried entity groups. The intermediate node groups contain a set of similar "intermediate' entities in the knowledge graph, whereas the queried entity group contains an entity in the query and semantically similar concepts to the queried entity that are contained in the knowledge graph.

### 3.4.3 KG-STRUCTURE GUIDED REASONING

In the previous sections, we pre-process the external structured knowledge source by constructing (1) groups of important concepts in Section 3.2, and any possible intermediate groups in Section 3.4.2. (2) possible connections between any two groups in Section 3.4.1. GIVE utilizes these non-parametric evidence $\mathbb{N}_x$ and $\mathbb{R}_x$ to inspire LLM conduct reasoning using its parametric knowledge, and formally build the inter-group knowledge set.

**Assigning relations with external evidence.** If there exists an edge on the external knowledge graph between a pair of nodes, we directly inherit the ground-truth relation from the original KG $G$. We consider all knowledge described on the external KG as ground-truth known facts. When we verbalize such edge in the prompt, we use affirmative tense to indicate such a fact is true with very high confidence.

**Veracity Extrapolation with internal knowledge.** The potential relations between node groups induced in Section 3.4.1 are crucial to inform us about the possible connections between nodes. The pre-training stage of LLM equips the model with rich factual knowledge from the unstructured corpus. It is important to concertize the relevant internal knowledge to affirm the model's decision or to reject wrong answers with explicit context. Thus, we prompt the LLM to assign a label to each potential relation among two node groups: "yes", "no", or "maybe". If the LLM yields "yes" for a certain relation, it indicates the model is confident that such a claim is factual.

It is important for the QA model to know not only what claims are highly likely to be factual but also the claims that are not going to hold or may not hold. This kind of counterfactual relation information prevents the model from hallucination. If the node pair does not contain a relation in the potential relation set indicated by a "no" answer returned by the LLM, we assign a reversed relation. If the model is not sure about a certain relation using its internal knowledge by answering "maybe", these connections are discarded as the LLM is not sure about its validity thus bears a higher chance of causing hallucination.

**Discovering open relations for novel connections.** To prevent the potential scope limitation of the relations presented in the knowledge graph, we additionally prompt the LLM to freely create a short phrase to describe the relation of a given node pair. Even two nodes can be connected through a novel relation that not presented in either question or the knowledge graph, the open relation discovery design keeps the flexibility of our proposed framework.

### 3.5 PROGRESSIVE ANSWER GENERATION

From the reasoning process presented in Section 3.4.3, we obtained three kinds of knowledge: (1) Affirmative knowledge set that contains all inner-group connection; all the potential cross-group connections that are labeled as "yes" by the LLM; and the cross-group connections that are built purely by the internal knowledge of LLM. We refer to this knowledge set as $\tilde{\mathcal{T}}_x^a(G)$. (2) Counterfactual knowledge set that contains all the potential relations that are labeled as "no" by LLM, which is referred to as $\tilde{\mathcal{T}}_x^c(G)$, (3) $\tilde{\mathcal{T}}_x^e(G)$, the ground-truth connection contained in the KG. To prevent hallucination, we adopt a progressive manner to generate the final answer by first giving only the

affirmative knowledge set. Then we ask the LLM to refine this answer by giving the full context of the previous step plus the counter-factual knowledge set. The final answer is generated by providing details of all previous context and the ground-truth knowledge contained in the original knowledge graph. Given generator $p_\gamma$ and the retrieved knowledge sets:

$$\text{GIVE}_a(y^a|x) \coloneqq p_\gamma(y^a|x, \tilde{\mathcal{T}}_x^a(G)) \tag{8}$$

$$\text{GIVE}_{a+c}(y^{a+c}|x, y^a) \coloneqq p_\gamma(y^{a+c}|x, \tilde{\mathcal{T}}_x^a(G), y^a, \tilde{\mathcal{T}}_x^c(G)) \tag{9}$$

$$\text{GIVE}_{a+c+e}(y^{a+c+e}|x, y^a, y^{a+c}) \coloneqq p_\gamma(y^{a+c+e}|x, \tilde{\mathcal{T}}_x^a(G), y^a, \tilde{\mathcal{T}}_x^c(G), y^{a+c}, \tilde{\mathcal{T}}_x^e(G)) \tag{10}$$

## 4 EXPERIMENTS

The experiments in this section are designed to answer the following research questions: (1) Is GIVE able to provide structured high-quality knowledge using very sparse external resources thus result in higher QA accuracy? We answer this in Section 4.2 by conducting experiments on various biomedical QA benchmarks using a small UMLS knowledge graph (Li et al., 2023). (2) Is GIVE robust to different sparsities of KG to retrieve useful information? To this end, we conduct experiments by randomly sampling different portions of triplets in ConceptNet (Speer et al., 2018), and use the resulted subgraphs to test on CommonsenseQA (Talmor et al., 2019) in Section 4.3 and truthfulQA (Lin et al., 2022) in Appendix D. (3) Is the performance of GIVE sensitive to the number of additional concepts in each group (which is the only hyper-parameter for our method)? We perform ablation studies in Section 4.4 to answer this. (4) On what kind of questions GIVE achieve the best performance? We give a detailed analysis in Appendix F.1 and conclude that GIVE improves the performance of LLM by achieving very high accuracy on the questions where the ratio of expert KG knowledge in the overall retrieved knowledge set is relatively high. (5)What are other factors that contribute to the performance of GIVE? We conducted additional ablation studies on a subset of each dataset in Appendix E.1, E.2, E.3 and E.4 on removing inner/inter group connections, the number of seeded examples, prompting strategies and encoder model size for entity group construction. (6) What is the detailed cost comparison between GIVE and the competing methods? We include a detailed discussion of GIVE's efficiency in terms of both running time and context length in Appendix F.2, in which we first prove that all factors that may influence GIVE's efficiency can be upper-bounded by a small number, and that GIVE is generally applicable to KGs of different sizes and sparcities in terms of accuracy and efficiency.

### 4.1 EXPERIMENTAL SETTINGS

#### 4.1.1 REASONING-RICH QA DATASETS

Since we aim to enable LLM to conduct faithful reasoning utilizing very limited external knowledge, the general KB-QA pairs with ground truth knowledge path does not align with the purpose of our experiments because they are designed to test effectiveness of information retrieval. Our experiments follow the principle that the involved KG should be related to the domain of the question, but not provide direct solution to the query, which creates an environment akin to human/expert deductive reasoning process, by following related high-level hints to solve the reasoning-intensive query. We quantify all KG and datasets included in our experiments in Appendix Section B.

**We focus on questions that are hard to answer without additional reasoning by ignoring any "gold" knowledge or context.** For PubmedQA (Jin et al., 2019), we challenge the competing methods by providing LLM with only the question statement and the retrieved facts, **not** any ground-truth gold context in which the answer is self-presented. Similarly, for BioASQ (Krithara et al., 2023), we extract all questions in Task 2B, 3B and 4B with both "ideal answer" (long answer) and "exact answer" (short answer). We ignore the long answer to test the accuracy on the short answer returned by each method. In the case of Processbank (Berant et al., 2014), we do **not** provide the ground-truth annotations by only giving the question statement and choices.

#### 4.1.2 COMPETING BASELINES AND BACKBONE LLMS

We compare the proposed GIVE framework against standard I/O prompting (Brown et al., 2020); CoT prompting (Wei et al., 2023); text-based RAG (Lewis et al., 2021), following the original setting to

use a DPR-based retriever (Karpukhin et al., 2020) on the verbalized triplets; ToG (Sun et al., 2024), which is the SOTA framework for retrieving structured information via KG-LLM interactions, and GraphRAG (Edge et al., 2024), which models not only the isolated pieces of external knowledge, but also the connections between the structured information unit. Note that GraphRAG is not designed to operate on large scale KGs and its performance is limited on smaller LLMs because of the difficulty in context understanding, so it is excluded in some comparisons. In biomedical reasoning tasks, we compare the performance of each competing method on GPT3.5-turbo, GPT-4, GPT4o-mini and Llama3.1-70B-Instruct, to prove the ability of GIVE in bridging the knowledge level between smaller and larger LLMs. For open-domain reasoning, we test on GPT3.5-turbo with ConceptNet with different edge ratios, to prove the robustness of GIVE in handling KGs with various sparcities and sizes. For each baseline methods, we provide the same set of randomly chosen k-shot examples, where k = 5 for "yes-no" datasets PubmedQA and BioASQ, and k = 10 for multiple-choice datasets ProcessBank and CommonsenseQA. To ensure fairness of comparison, for CoT, RAG, ToG and GraphRAG, we also provide the correct reasoning process for each of the given examples. We provide the full details of prompts used for each baseline in Appendix G.

## 4.2 BIOMEDICAL REASONING ON SMALL UMLS

Table 1: Performance on Biomedical QA in accuracy (%) using GPT series backbone models. Retrieval-based methods are given access to a sparse UMLS KG (Li et al., 2023). Each method is provided with the same few-shot examples. We highlight in green the largest performance improvement of the proposed GIVE framework, compared to the (1) best of {I/O prompting (Brown et al., 2020), CoT prompting (Wei et al., 2023)}, (2) text-based RAG (Lewis et al., 2021), (3) best of {ToG (Sun et al., 2024), GraphRAG (Edge et al., 2024)}.

| # | Method/Dataset | GPT3.5-turbo | | | GPT4 | | | GPT4o-mini | | |
|---|---|---|---|---|---|---|---|---|---|---|
| | | PubMedQA | BioASQ | ProcessBank | PubMedQA | BioASQ | ProcessBank | PubMedQA | BioASQ | ProcessBank |
| | | *Internal knowledge reasoning* | | | | | | | | |
| 1 | I/O prompt | 46.2 | 43.5 | 67.3 | 42.2 | 88.2 | 64.8 | 23.4 | 88.7 | 79.4 |
| 2 | CoT prompt | 48.6 | 63.5 | 70.9 | 37.8 | 80.4 | 59.3 | 23.8 | 79.3 | 81.4 |
| | | *External knowledge (text) reasoning* | | | | | | | | |
| 3 | RAG | 13.4 | 40.9 | 67.3 | 26.4 | 24.3 | 78.9 | 15.2 | 16.3 | 84.9 |
| | | *External knowledge (KG) reasoning* | | | | | | | | |
| 4 | ToG | 17.6 | 18.0 | 66.8 | 19.1 | 15.4 | 81.4 | 21.8 | 10.1 | 84.4 |
| 5 | GraphRAG | 23.4 | 10.3 | 71.3 | 26.5 | 11.2 | 80.9 | 22.6 | 10.1 | 84.9 |
| | | *Structured reasoning with internal and external knowledge(Our method)* | | | | | | | | |
| 5 | GIVE$_a$ | 44.4 | 82.6 | 72.9 | 50.0 | **90.0** | 82.7 | 26.0 | **89.5** | 85.9 |
| 6 | GIVE$_{a+c}$ | 49.8 | 86.1 | **73.9** | **50.2** | 80.6 | **83.3** | **27.4** | 81.9 | **87.4** |
| 7 | GIVE$_{a+c+e}$ | **53.6** | **88.2** | 73.4 | 43.4 | 87.8 | 82.7 | 27.2 | 81.9 | 86.9 |
| 8 | Best Gain(+%) | 5/40.2/30.2 | 24.7/47.3/70.2 | 3/6.6/2.6 | 8/23.8/23.7 | 1.8/65.7/74.6 | 18.5/4.4/1.9 | 3.6/12.2/4.8 | 0.8/73.2/79.4 | 8/2.5/2.5 |

**GIVE enables smaller-sized LLMs to achieve better performance than the most advanced models with very limited external knowledge in scientific domains where both training on and retrieving from inclusive knowledge base is hard.** Our first observation is that GIVE consistently achieves the best performance among all reasoning and retrieval-based baselines. Especially, GIVE enables GPT3.5-turbo to surpass GPT4 uniformly on biomedical reasoning tasks. For example, on BioASQ, GIVE offers GPT3.5-turbo an accuracy boost of 44.7%, resulting in an accuracy that is 11.4% higher than GPT4. If we compare the results of I/O prompt across different models, we see training on these scientific domains proved to be very hard, especially in case of PubmedQA (Jin et al., 2019) and ProcessBank (Berant et al., 2014). On the other hand, retrieving precise knowledge from the limited resources is also infeasible, we see this from the performance of RAG (Lewis et al., 2021). In this case, GIVE successfully combines the training time and inference time knowledge by using a sparse KG of only 135 nodes, without any additional training cost.

**GIVE is flexible to operate on LLMs with different sizes and levels of internal knowledge.** Comparing the results in Table 1 and Table 2, we observe that GIVE is able to boost the reasoning ability of LLMs with different sizes (GPT4 > GPT3.5T > Llama3.1 > GPT4o-mini). Furthermore, there are two important factors for the performance increase offered by GIVE compared to I/O prompt or CoT, the model size and whether or not it has enough internal knowledge to answer the questions. Specifically, the larger the backbone model is, the higher accuracy in-

crease GIVE is able to offer. We see this by comparing the performance gain on PubMedQA. In case of BioASQ and Processbank, I/O prompting already achieves an accuracy of around 90%, GIVE is still able to increase the performance of the model using very limited external evidence.

**GIVE effectively prevents hallucination introduced by the sparse knowledge source.** We also see that the advantage of GIVE is more significant compared to the retrieval-based methods that tries to directly use the knowledge retrieved from the sparse external knowledge source. This is because the triplets retrieved by DPR (Karpukhin et al., 2020) and ToG (Sun et al., 2024) are low-quality and the model is influenced a lot by these irrelevant information. We see this from Figure 12 and 11. This is particularly the case when they operate on a strong model that already has rich internal knowledge (GPT4/4o-mini on BioASQ). It turns out to be an important problem to prevent such hallucination when deploying LLMs in knowledge-intensive domains with limited external resource. GIVE provides a low-cost solution to this challenging scenario, for it is not only robust to hal-

Table 2: Performance on Biomedical QA using backbone LLM Llama-3.1.

| # | Method/Dataset | Meta-Llama-3.1-70B-Instruct | | |
|---|---|---|---|---|
| | | PubMedQA | BioASQ | ProcessBank |
| | | *Internal knowledge reasoning* | | |
| 1 | I/O prompt | 48.0 | 91.0 | 85.4 |
| 2 | CoT prompt | 50.4 | 91.3 | 84.3 |
| | | *External knowledge (text) reasoning* | | |
| 3 | RAG | 49.8 | 45.4 | 84.4 |
| | | *External knowledge (KG) reasoning* | | |
| 4 | ToG | 38.4 | 31.0 | 85.9 |
| | | *Internal and external knowledge reasoning* | | |
| 5 | GIVE$_a$ | 56.0 | 91.7 | 86.4 |
| 5 | GIVE$_{a+c}$ | **56.2** | 91.7 | **86.9** |
| 5 | GIVE$_{a+c+e}$ | 56.0 | **92.6** | 86.4 |
| 6 | Best Gain(+%) | 5.8/6.3/17.8 | 1.3/47.2/61.6 | 1.5/2.5/1 |

lucination introduced by the irrelevant knowledge, but also capable of improving the performance, even using very limited expert domain knowledge.

**GIVE$_{a+c+e}$ achieves the most consistent performance compared to GIVE$_a$ and GIVE$_{a+c}$.** Since GIVE$_{a+c+e}$ takes use of all the generated knowledge as illustrated in Section 3.5. This implies the contour-factual knowledge retrieval process we propose in Section 3.4.3 provides useful additional information to guide reasoning, also underlines the importance of properly incorporating sparse KG information in knowledge-intensive QA tasks. This is further discussed in Appendix Section F.1.

## 4.3 COMMONSENSE REASONING ON SPARSE/HALF/FULL CONCEPTNET

**GIVE is effective in retrieving information from both sparse and dense KG.** Regarding the ability of GIVE to generate useful information from both sparse and complete KG, the conclusion is definite. As we can see from Table 3, on the full ConceptNet, GIVE offers 3.4% and 4.9% accuracy increase compared to RAG (Lewis et al., 2021) and ToG (Sun et al., 2024). Retrieving information from very dense KG on a specific domain also poses significant challenge because of the large number of similar entities and triplets. The results proved the robustness of GIVE to generate useful information, thus prompting structured reasoning for LLMs using different sparcities of external knowledge source.

## 4.4 ABLATION STUDY

**GIVE achieves the best performance using only small number of additional KG entities.** The key parameter of GIVE is the number of additional KG entities we introduced to each

Table 3: Performance on CommonsenseQA (Val set) in accuracy(%) on GPT3.5-turbo. Retrieval-based methods are given access to a sub-graph of ConceptNet (Speer et al., 2018) with different portions of randomly sampled triplets. We highlight the performance improvement of GIVE compared to the (1)best of {I/O prompting(Brown et al., 2020), CoT prompting(Wei et al., 2023)}, (2)RAG (Lewis et al., 2021), (3)ToG (Sun et al., 2024).

| # | Method / % of triplets | Commonsense QA | | |
|---|---|---|---|---|
| | | 10% | 50% | 100% (Full) |
| | | *Internal knowledge reasoning* | | |
| 1 | I/O prompt | | 71.8 | |
| 2 | CoT prompt | | 72.2 | |
| | | *External knowledge (text) reasoning* | | |
| 3 | RAG | 70.4 | 70.6 | 71.3 |
| | | *External knowledge (KG) reasoning* | | |
| 4 | ToG | 69.7 | 71.2 | 69.8 |
| | | *Internal and external knowledge reasoning* | | |
| 5 | GIVE$_a$ | 73.3 | 73.6 | 74.2 |
| 6 | GIVE$_{a+c}$ | 73.4 | 73.6 | 74.2 |
| 7 | GIVE$_{a+c+KG}$ | **73.5** | **73.8** | **74.7** |
| 8 | Best Gain(+%) | 1.3/3.1/3.8 | 1.6/3.2/2.6 | 2.5/3.4/4.9 |

concept group. To study how that influences the performance of GIVE, we conduct experiments on biomedical reasoning with GPT3.5-turbo, using number of KG entities from 0 to 3. As shown

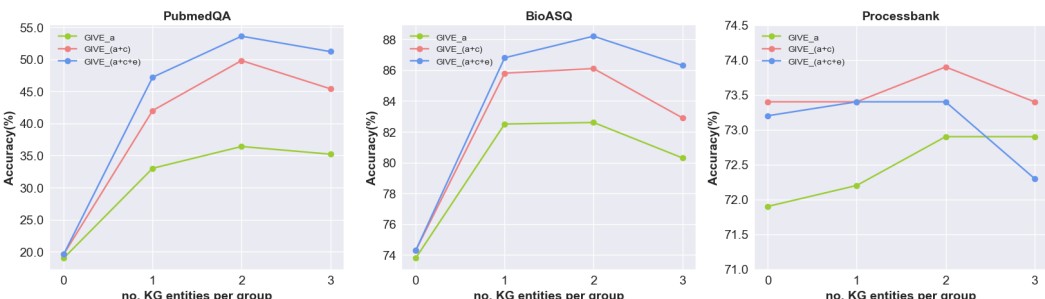

Figure 4: Performance of GIVE with different numbers of entities per group

in Figure 4, the performance of GIVE improves first with increasing number of KG entities per group from 0 to 2, and decreases when we increase it to 3 and the observation is uniform across all datasets. This is because GIVE first enables LLM to conduct structured reasoning using the additional information. Since we are using a sparse KG with only 135 nodes, when k is greater than 2, it is very likely to introduce entities that is not directly related to the queried concepts, and thereby causes hallucination.

**Get inspired, do not recite.** The performance jump when increasing the number of KG entities from 0 to 1 proves the effectiveness of the "Graph Inspiration" process we proposed in Section 3.4.3, by introducing additional related concepts from external source and "inspire" the LLM to conduct divergent reasoning using these external clues. This points out that the ability of divergent thinking of LLMs may have long been ignored, as we have been focusing on retrieving the gold knowledge for the model to "recite". Instead, further studies should be conducted on how to utilize the external knowledge as a high-level clue to "inspire" LLMs conduct reasoning, rather than a "long-answer" style gold context. This is especially the case when we deploy LLMs in scientific domains where both training on and retrieving from inclusive knowledge are infeasible.

**Limitation statement:** It remains a heuristic on how to eliminate hallucination caused by in-accurate knowledge GIVE introduced, as there is no performance guarantee on the LLM's ability to prune out the correct potential knowledge from the wrong. In fact, it is related to the size of the LLM and how extensively it has been trained on the specific domain knowledge. Regarding the complexity of GIVE, suppose we have m entity groups and each group has n concepts, between two entity groups there r candidate relations. The inner-group connections (Section 3.3) takes $\mathcal{O}(mn)$ LLM calls. For inter-group connections (Section 3.4.3), the number of LLM calls needed equal to the number of generalized potential connections, which is $\mathcal{O}(rm^2n^2)$. However, as shown in Section 4.4, GIVE achieves best performance when n = 1 or 2. In Appendix F.2 we further prove that (1) average value for m is around 3 or 4 for all datasets; (2) average value for r is upper-bounded by 4 for all datasets; (3) both running time and context length of GIVE remain reasonable when we increase the size or sparsity of the KG, and are far from the limits of the widely-used LLMs. (4) the large amount of knowledge retrieved by GIVE is of high-quality and is important to improve the model's performance. (5) complexity of GIVE can be further reduced by: divide-and-conquer prune the knowledge in batches; include summarization agents to reduce the length of the knowledge.

## 5 CONCLUSION

We propose Graph Inspired Veracity Extrapolation (GIVE), a knowledge extrapolation framework for structured reasoning of LLM on sparse knowledge graphs. GIVE neither focuses on explicit information retrieval, nor relies on improving the internal reasoning ability of LLMs by appending triggering statements to the query. It utilizes the high-level thinking processes mined in sparse knowledge graphs to combine both approaches. It retrieves the most relevant information in the knowledge base and, at the same time, inspires LLM to exploit its internal knowledge by conducting structured reasoning and knowledge extrapolation. GIVE provides substantial amount of performance increase in different QA tasks using KGs of various sizes and sparcities, and mitigates the hallucination issue of retrieval-based methods on non-inclusive knowledge source. It sheds light on the potential of LLM to conduct divergent reasoning using very limited external clues.

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

## A    ALGORITHM FOR GIVE

We summarize the comprehensive procedure of GIVE and present its detailed algorithm in Algorithm 1

---

**Algorithm 1:** GIVE

---

**Input:** Entity groups $\mathbb{N}_x$; Possible relations between two entity groups $\mathbb{R}_x$; Knowledge Graph G

**Output:** $\tilde{\mathcal{T}}_x(G)$, the approximated gold knowledge set that helps to solve query x

1  $\tilde{\mathcal{T}}_x^a(G) \leftarrow \varnothing$
2  $\tilde{\mathcal{T}}_x^c(G) \leftarrow \varnothing$
3  $\tilde{\mathcal{T}}_x^e(G) \leftarrow \varnothing$
4  **for** *all queried entity $e_x^i$ and their corresponding relevant concepts $y_x^j \in N_x^i$* **do**
5      // build inner-group connections
    $(e_x^i, y_x^j) \rightarrow \text{LLM} \rightarrow (e_x^i, r_x^{ij}, y_x^j)$
6      $\tilde{\mathcal{T}}_x^a(G) \leftarrow \tilde{\mathcal{T}}_x^a(G) \cup \{(e_x^i, r_x^{ij}, y_x^j)\}$
7  **for** *$(N_x^i, N_x^j)$ pairs in $\mathbb{N}_x \times \mathbb{N}_x$* **do**
8      // build inter-group connections
    retrieve all triplets $\tilde{T}_x^e(G^{ij}) \in \mathcal{E}_G$ connecting any node in $N_x^i$ and any node in $N_x^j$
9      $\tilde{\mathcal{T}}_x^e(G) = \tilde{\mathcal{T}}_x^e(G) \cup \tilde{\mathcal{T}}_x^e(G^{ij})$
10     $R_G^{ij} \leftarrow$ set of relation types in $\tilde{\mathcal{T}}_x^e(G^{ij})$ $R_x^{ij} \leftarrow R_G^{ij} \cup R_x$
11     **for** *all triplets $(n_x^i, r_x^{ij}, n_x^j)$ in $(N_x^i \times R_x^{ij} \times N_x^j)$* **do**
12         $(n_x^i, r_x^{ij}, n_x^j) \rightarrow \text{LLM} \rightarrow$ yes,no or maybe
13         **if** *yes* **then**
14             $\tilde{\mathcal{T}}_x^a(G) = \tilde{\mathcal{T}}_x^a(G) \cup (n_x^i, r_x^{ij}, n_x^j)$
15         **if** *no* **then**
16             $\tilde{\mathcal{T}}_x^c(G) = \tilde{\mathcal{T}}_x^c(G) \cup (n_x^i, \text{not } r_x^{ij}, n_x^j)$
17 **return** $\tilde{\mathcal{T}}_x^a(G), \tilde{\mathcal{T}}_x^c(G), \tilde{\mathcal{T}}_x^e(G)$

---

## B    DIFFERENCE BETWEEN GIVE AND RETRIEVAL AUGMENTED GENERATION (RAG)

**Retrieval-Augmented Generation.** An LLM $p_\theta(y|x)$ generates the probability distribution of output $y$ given an input $x$. RAG-based systems (Lewis et al., 2021) enhance the capability of language models in knowledge-intensive tasks by leveraging a retriever-generator framework. The retriever model is denoted as $p_\eta(z|x, D)$, where $x$ is an input query, $D$ is a comprehensive knowledge base, and it generates knowledge distribution over the given input and knowledge base. The generator $p_\theta(y|z, x)$ then autoregressively generates the output sequence based on the retrieved knowledge $z$ and the input context $x$. The likelihood of generating an output sequence $y = y_{1:N}$ can be estimated as:

$$p(y|x, D) := p_\eta(z|x, D)p_\theta(y|x, z) \tag{11}$$

Recent studies (Edge et al., 2024) proposes to improve RAG by empirically proving that it is beneficial to model the internal connections between information units, rather than feed the model with independent knowledge. These methods, and the algorithms to tackle KBQA, focus on information retrieval and they aim to provide the most accurate external knowledge contained in the documents. GIVE provides another axis for solving problems using external knowledge, is to combine the limited resource and the parametric knowledge to generate high quality response. We are convinced that this is important as we move forward the test time scaling era of deploying artificial intelligence.

## C    DATASET DETAILS

Table 2: Summary of Dataset statistics.

| Task | KG | $|\mathcal{V}|$ | $|\mathcal{E}|$ | Datasets | QA Type |
|---|---|---|---|---|---|
| Biomedical Reasoning | UMLS (Li et al., 2023) | 135 | 5,877 | PubmedQA (Jin et al., 2019)
BioASQ (Krithara et al., 2023)
ProcessBank (Berant et al., 2014) | Yes-No
Yes-No
Multiple-Choice |
| Commonsense Reasoning | 10% ConceptNet (Speer et al., 2018)
50% ConceptNet (Speer et al., 2018)
Full ConceptNet (Speer et al., 2018) | 223,863
607,483
844,158 | 208,510
1,042,550
2,085,099 | CommonsenseQA (Talmor et al., 2019) | Multiple-Choice |
| Opendomain Reasoning | 10% ConceptNet (Speer et al., 2018) | 223,863 | 208,510 | TruthfulQA (Lin et al., 2022) | Text Generation |

We quantify the statistics of all knowledge graph used in our experiments and the task types of all QA datasets in Table 2. We include different types of KGs, to showcase the robustness of GIVE to handle different types of external resources. UMLS (Li et al., 2023) used in bio-medical reasoning tasks is a small but dense knowledge graph, whereas in Commonsense reasoning, we use ConceptNet (Speer et al., 2018) with different edge ratios. We also include different types of QA tasks including "Yes-No", "Multiple-Choice", and "Text Generation".

# D  ANSWER GENERATION ON TRUTHFULQA

In addition to Section 4.2 and 4.3, we include additional experiments to test GIVE's performance on open-domain QA dataset. We compare the performance of each competing method using TruthfulQA(Lin et al., 2022) on answer generation task. For retrieval-based method (RAG, ToG, GIVE), we use the 10% ratio ConceptNet to challange their ability to retrieve information from sparse KG in open-domain answer generation. We use GPT4o to compare the win rate and the results are reported in Figure 5. Specifically, for each question, we provide GPT4o with the best answer, and the answer of each method, ask it to score each answer by their semantic similarity with the ground truth best answer, and select the one with highest score as winning answer.

**GIVE is effective in both domain-specific and open-domain reasoning tasks.** We observe from Figure 5 that GIVE achieves the best score in totally 50.3% of the questions in TruthfulQA (Lin et al., 2022). The results proved that GIVE is effective in both scientific domain reasoning and open-domain question answering.

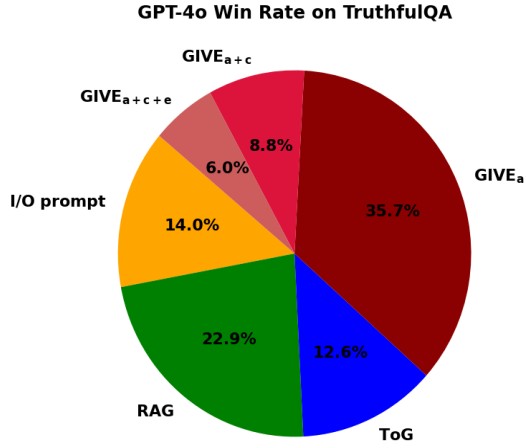

Figure 5: GPT4o win rate on TruthfulQA (Lin et al., 2022) in %.

It is also noticed that $GIVE_a$ achieves the most best answers compared to $GIVE_{a+c}$ and $GIVE_{a+c+e}$, this is because this dataset is designed to test the truthfulness of the answer, not reasoning ability of the model. For example, "What is the original meaning of SOS?" To answer this question, no counter-factual knowledge will help. When humans are asked this question, we do not reason by listing a number of "SOS does not mean..." statements. In case of LLM, providing them may even cause hallucination. We conclude that to use GIVE in such open-domain datasets, we should rely more on the affirmative knowledge set. In other words, answers from $GIVE_a$ should be preferred.

# E  ADDITIONAL ABLATION STUDIES

In addition to Section 4.4, we conduct more detailed ablation studies for GIVE to study the robustness of the proposed method and other factors that may influence its performance. All experiments in this Section are based on 50 randomly generated examples for each dataset, whereas in Section 4.4, we study the influence of different numbers of entities per group on the full dataset.

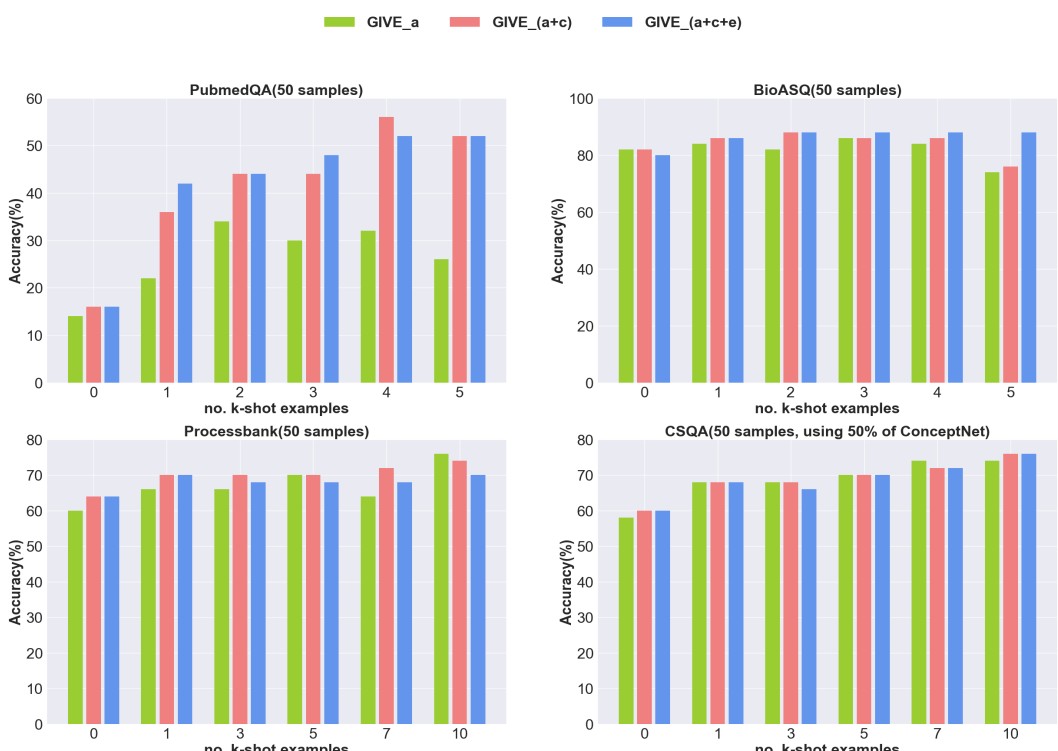

Figure 7: Sensitivity analysis of GIVE on different number of seeded examples.

### E.1 INNER-GROUP AND INTER-GROUP CONNECTIONS

We conduct experiments to test the importance of each type of connections we build in Section 3.4.3. We test the performance of GIVE using knowledge induced from (1) only inner-group connections in Section 3.3. (2) only inter-group connections in 3.4. (3) both inner-group and inter-group connections. The results are presented in Figure 6.

**Both inner-group and inter-group connections are necessary of GIVE, but inter-group connections contribute the most to the performance.** We observe that GIVE achieves the best performance when we incorporate both inner-group and inter-group connections. However, we observe a performance leap of GIVE when we add the inter-group connections. This is because the inner-group connections are added to bridge the query and the knowledge graph concepts, whereas the inter-group connections produce the necessary knowledge to bridge the different entity groups thus prompt a faithful reasoning process to solve the query.

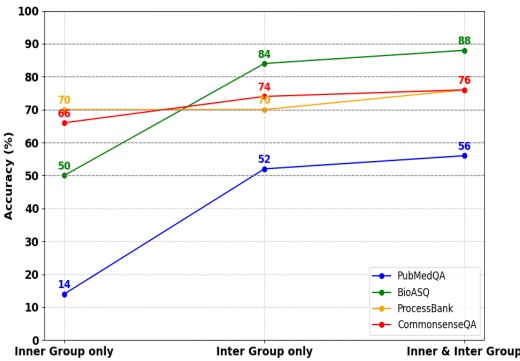

Figure 6: Performance comparison of GIVE using only inner-group connections, inter-group connections, both inner/inter group connections on 100 random samples from each dataset.

### E.2 NUMBER OF SEEDED EXAMPLES

To better understand how difficult it is for LLMs to get the generalized ability to adopt the knowledge generated by GIVE to build the structured reasoning chain, we study the performance of GIVE by providing different number of examples in the prompt. For yes-no datasets PubmedQA (Jin et al.,

2019) and BioASQ (Krithara et al., 2023), we randomly choose k of {0, 1, 2, 3, 4, 5} examples. For multiple-choice datasets Processbank (Berant et al., 2014) and CSQA (Talmor et al., 2019), we choose k of {0, 1, 3, 5, 7, 10}. The results are presented in Figure 7.

We observe that although the performance of GIVE increases as we give more seeded examples in the prompt, the only one large performance upgrade happens when we increase the number of examples from 0 to 1. This implies that GIVE is a generalizable framework for the LLM to easily adopt. The high performance of GIVE does not rely on large number of examples, but stems from the high quality of the synthetic data it generates.

### E.3 DIFFERENT WAYS OF PROMPTING

We perform additional experiments in this sub-section to study how different prompting strategies influence the performance of GIVE. We verbalize the retrieved knowledge and prompt them in the form of triplets and text, and the results are presented in in Table 3. We notice that in most cases, prompting the knowledge in triplets yields to higher accuracy than prompting knowledge in text. This is because the structure of triplets naturally provides an easier way for the LLM to connect the related entities and build faithful logical chain to solve the question. However, for text-based information, additional analyzing step is needed to understand the text before it links the useful information together, which is a difficult task for reasoning-intensive queries where the volumn of additional knowledge is high.

Table 3: Performance of GIVE using different prompting methods on 50 randomly chosen examples for each dataset. We highlight in green the better-performed prompting method and the performance difference.

| # Prompting Method / dataset | GPT3.5-turbo | | | |
| | PubmedQA | BioASQ | Processbank | CSQA |
|---|---|---|---|---|
| | **GIVE$_a$** | | | |
| 1  Triplet prompt | 32 | 86 | 76 | 74 |
| 2  Text prompt | 46 | 86 | 74 | 62 |
| | **GIVE$_{a+c}$** | | | |
| 1  Triplet prompt | 56 | 86 | 74 | 76 |
| 2  Text prompt | 54 | 84 | 74 | 70 |
| | **GIVE$_{a+c+e}$** | | | |
| 1  Triplet prompt | 52 | 88 | 70 | 76 |
| 2  Text prompt | 54 | 84 | 72 | 68 |

### E.4 ENCODING MODEL SIZE

In Section 3.2, we employ SentenceTransformer as encoder model to measure the text similarities for entity group construction. We investigate the impacts of using different sizes on the performance of GIVE, and demonstrate the results in Table 4. We see that although larger size encoder models achieve better sentence embedding or performance semantic search performance, small to middle size encoders tend to perform more consistently on all datasets. For the best-performing GIVE$_{a+c+e}$, the 80M encoder (all-MiniLM-L6-v2) achieves 8% higher accuracy than the 420M one (all-mpnet-base-v2). The results show that larger size encoders do not necessarily better measure text similarity between specific domain terms. On the other hand, the performance of GIVE does not rely on the size of the models employed, which enhances the efficiency of GIVE.

## F  DETAILED ANALYSIS OF GIVE

### F.1  WHAT MAKES GOOD "INSPIRATIONS"?

We noticed that the accuracy improvement of GIVE alternates across different QA datasets, to carefully examine what makes the different capabilities of GIVE in boosting LLM's performance, we randomly sample 50 questions from PubemdQA (Jin et al., 2019), BioASQ (Krithara et al., 2023), ProcessBank (Berant et al., 2014) and CSQA (Talmor et al., 2019).

Specifically, From Table 1 and Table 3, we found that when vanilla LLM does not have enough internal knowledge (I/O prompting gets poor performance), the accuracy improvement achieved by GIVE has the trend of BioASQ > PubmedQA > Processbank > CSQA. **The different performance of GIVE stems from the ratio of expert KG knowledge in the whole retrieved knowledge set.** To better see this, we define expert ratio for a query x to be $\frac{|\tilde{\mathcal{T}}(\mathcal{G})|}{|\tilde{\mathcal{T}}(\mathcal{G})|+|\tilde{\mathcal{T}}(\mathcal{G})|+|\tilde{\mathcal{T}}(\mathcal{G})|}$, where $\tilde{T}_x(G)^e$, $\tilde{T}_x(G)^a$ are the set of expert KG knowledge, affirmative knowledge and counter-

Table 4: Performance of GIVE using GPT3.5-turbo and encoding SentenceTransformers of different sizes to search for relevant entities to build entity group (Section 3.2). Results are based on 50 randomly generated samples for each dataset. We highlight the results from the best performing model in green.

| # Encoding model(size) / dataset | GPT3.5-turbo | | | |
|---|---|---|---|---|
| | PubmedQA | BioASQ | Processbank | CSQA |
| **GIVE$_a$** | | | | |
| 1 paraphrase-albert-small-v2(43M) | 44 | 84 | 74 | 68 |
| 2 all-MiniLM-L6-v2(80M) | 32 | 86 | 76 | 74 |
| 3 all-MiniLM-L12-v2(120M) | 24 | 80 | 62 | 72 |
| 4 all-mpnet-base-v2(420M) | 38 | 88 | 66 | 64 |
| **GIVE$_{a+c}$** | | | | |
| 1 paraphrase-albert-small-v2(43M) | 54 | 82 | 76 | 70 |
| 2 all-MiniLM-L6-v2(80M) | 56 | 86 | 74 | 76 |
| 3 all-MiniLM-L12-v2(120M) | 52 | 82 | 62 | 70 |
| 4 all-mpnet-base-v2(420M) | 52 | 86 | 62 | 64 |
| **GIVE$_{a+c+e}$** | | | | |
| 1 paraphrase-albert-small-v2(43M) | 52 | 84 | 76 | 72 |
| 2 all-MiniLM-L6-v2(80M) | 52 | 88 | 70 | 76 |
| 3 all-MiniLM-L12-v2(120M) | 54 | 82 | 62 | 70 |
| 4 all-mpnet-base-v2(420M) | 52 | 88 | 60 | 64 |

factual knowledge we retrieved from Section 3.4.3. We then calculate the average expert ratio of 50 randomly chosen samples for each dataset and demonstrate the relationship between the average expert ratio and the best accuracy gain of GIVE compared to I/O prompting in Figure 8.

**There is a positive correlation between the performance of GIVE and the ratio of expert KG knowledge.** The reason behind this is with the larger number of seeded expert triplets, GIVE would have more concrete candidate relations and related entities for the LLM to conduct divergent thinking in the proposed "inspiration" process. The quality of the synthetic knowledge depends on the number of seeded expert KG knowledge provided. In the case that there are only few KG knowledge (for CSQA on the 50%-triplet Conceptnet), most retrieved knowledge are based only on LLM's internal knowledge to decide openly what the relationship is between two concepts. When the give KG is rich in information, the ground truth triplets provide a high-quality "supervise" for the "inspiration" process to "hint" the model what kind of relationship may exist between the entities. To further backup this statement, we divide these 50 randomly sampled questions from PubemdQA, BioASQ and Processbank into sub-groups according to their expert ratio, and we calculate the average accuracy for each sub-group. The results are demonstrated in Figure 9. We get the uniform conclusion that on every dataset that expert guidance

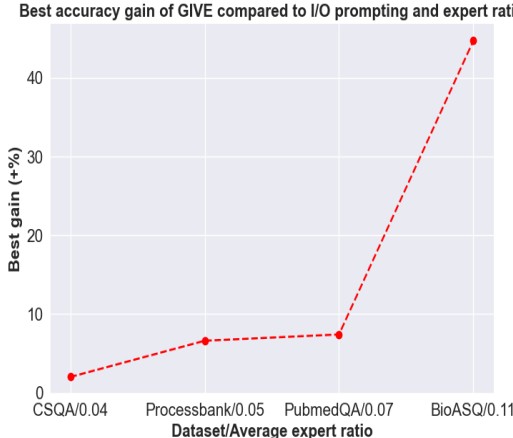

Figure 8: Best accuracy gain of GIVE and expert ratios for each dataset on 50 randomly chosen questions. For CSQA, we report the results using 50%-triplet version of ConceptNet.

(KG knowledge) is available, GIVE gets very high performance on the questions with high expert ratio. On the questions that nearly purely relies on the internal knowledge of LLM, the performance of GIVE is much degraded. It turns out that **neither external knowledge or internal knowledge itself is able to solve knowledge-intensive tasks, efforts must be made to fill this gap, and GIVE is designed for this.**

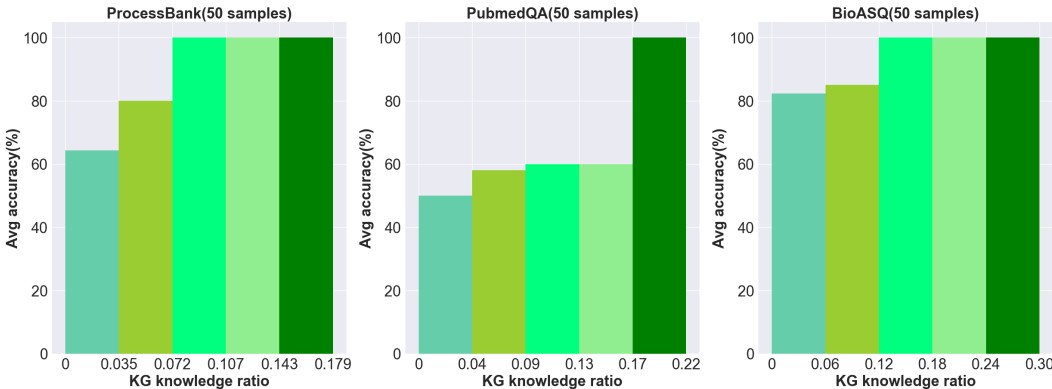

Figure 9: Accuracy achieved by GIVE for questions with different expert ratio (KG knowledge ratio), calculated by $\frac{|\tilde{\mathcal{T}}(\mathcal{G})|}{|\tilde{\mathcal{T}}(\mathcal{G})|+|\tilde{\mathcal{T}}(\mathcal{G})|+|\tilde{\mathcal{T}}(\mathcal{G})|}$, where $\tilde{T}_x(G)^e, \tilde{T}_x(G)^a$ are the set of expert KG knowledge, affirmative knowledge and counter-factual knowledge we retrieved from Section 3.4.3.

## F.2 EFFICIENCY OF GIVE

In Section 4.4 we discussed the efficiency of GIVE and concluded that the key factors that influence the number of LLM calls required by GIVE is the number of entity groups detected for the query and the number of candidate relations between each pair of entity groups. To conduct a more detailed study on the scale of them, we run GIVE 5 times on 50 randomly selected questions for each of the datasets we included in Section 4.2 and 4.3, and we report the average number of entity groups, average number of candidate relations to connect two entity groups, and average percentage of questions that requires intermediate entity groups (3.4.2) for multi-step reasoning. The results are presented in Figure 10.

We observe that on average, GIVE requires around 3 entities groups for each question in the Biomedical datasets (PubmedQA, BioASQ, Processbank), between each datasets, there could be 1 to 6 candidate relations. For commonsenseQA, 4 entity groups on average are detected because the dataset has 5 candidate options, between each pair of entity groups, only 1 candidate relation is detected in general. We also notice that 60% of the questions in PubMedQA requires intermediate group. That is the reason why PubMedQA tends to need more entity groups than BioASQ as a "yes-no" QA dataset. This implies one of the potential method to improve efficiency of GIVE is to disable intermediate group detection. On the other hand, we can use the LLM to prune the candidate connections in batches, which means in Section 3.4.3, instead of asking LLM "yes" or "no" for each potential connection, we can prompt the LLM with a set k of relations and let it select out which ones are true of false, which will devide the total number of LLM calls by the factor of k for GIVE.

We further conduct experiments to compare the efficiency of GIVE with RAG (Lewis et al., 2021) and ToG (Sun et al., 2024), in terms of running time and context length, for every experiment setting we include in Section 4, the results are presented in Table 5.

**The computational cost of GIVE remains reasonable as we increase the size and density of the KG.** (1) In terms of running time, when we increase the density (number of edges) to ×5 or ×10 on ConceptNet (Speer et al., 2018), we see a sub-linear running time increase for GIVE. Even with n=2, GIVE achieves shorter or comparable running time with ToG (Sun et al., 2024). This proves the $\mathcal{O}(\ )$ running time of GIVE, as pointed out in Section 4.4. When we increase the density of the KG, the only factor that will change is $r$, which is the number of relations between two entity groups, the increase of which is strictly upper-bounded by the increase of total number of edges

Table 5: Efficiency comparison between GIVE and RAG (Lewis et al., 2021), ToG (Sun et al., 2024). On each dataset, we run every method on 100 randomly selected questions, we report the average running time in seconds, context length in number of words, and accuracy in %. For RAG, we retrieved top 10 knowledge and for ToG, we use search depth=5 to maximize their performance, the settings are the same with experiments in Section 4. For GIVE, we report the results for both n=1 and n=2, where n is the number of additional KG concepts per group. $\tilde{\mathcal{T}}_x^a(G)$, $\tilde{\mathcal{T}}_x^c(G)$, $\tilde{\mathcal{T}}_x^e(G)$ are the retrieved affirmative knowledge set, counter-factual knowledge set and expert KG knowledge set, respectively.

| # Method/Dataset | time(s) | Context Length (# words) | | | Acc(%) | | |
|---|---|---|---|---|---|---|---|
| | | $\tilde{\mathcal{T}}_x^a(G)$ | $\tilde{\mathcal{T}}_x^c(G)$ | $\tilde{\mathcal{T}}_x^e(G)$ | $\text{GIVE}_a$ | $\text{GIVE}_{a+c}$ | $\text{GIVE}_{a+c+e}$ |
| *PubmedQA on UMLS* | | | | | | | |
| 1 RAG | 2.7 | | 64.7 | | | 14 | |
| 2 ToG | 15.9 | | 73.8 | | | 16 | |
| 3 $\text{GIVE}_{n=1}$ | 33.6 | 192.5 | 105.9 | 16.1 | 35 | 45 | 45 |
| 4 $\text{GIVE}_{n=2}$ | 103.8 | 456.3 | 486.1 | 57.6 | 41 | **48** | **48** |
| *BioASQ on UMLS* | | | | | | | |
| 1 RAG | 2.8 | | 66.5 | | | 43 | |
| 2 ToG | 10.3 | | 42.7 | | | 17 | |
| 3 $\text{GIVE}_{n=1}$ | 15.3 | 83.6 | 33.8 | 9.0 | 79 | 81 | 83 |
| 4 $\text{GIVE}_{n=2}$ | 45.3 | 205.5 | 176.5 | 32.1 | 80 | 84 | **90** |
| *Processbank on UMLS* | | | | | | | |
| 1 RAG | 2.8 | | 61.7 | | | 68 | |
| 2 ToG | 15.6 | | 54.2 | | | 60 | |
| 3 $\text{GIVE}_{n=1}$ | 35.2 | 151.2 | 226.9 | 7.5 | 67 | 68 | 68 |
| 4 $\text{GIVE}_{n=2}$ | 93.8 | 354.0 | 649.5 | 28.9 | 71 | 71 | **72** |
| *CSQA on 10% ConceptNet* | | | | | | | |
| 1 RAG | 0.6 | | 30 | | | 64 | |
| 2 ToG | 39.3 | | 106.6 | | | 67 | |
| 3 $\text{GIVE}_{n=1}$ | 26.5 | 41.1 | 38.4 | 0.1 | 70 | **71** | **71** |
| 4 $\text{GIVE}_{n=2}$ | 36.3 | 93.2 | 83.3 | 0.2 | 70 | 70 | 68 |
| *CSQA on 50% ConceptNet* | | | | | | | |
| 1 RAG | 1.1 | | 30 | | | 66 | |
| 2 ToG | 102.2 | | 217.0 | | | 67 | |
| 3 $\text{GIVE}_{n=1}$ | 74.0 | 39.9 | 43.5 | 0.2 | 69 | 64 | 65 |
| 4 $\text{GIVE}_{n=2}$ | 82.0 | 86.9 | 91.6 | 0.5 | 73 | **76** | 75 |
| *CSQA on full ConceptNet* | | | | | | | |
| 1 RAG | 1.7 | | 30 | | | 69 | |
| 2 ToG | 125.2 | | 213.7 | | | 63 | |
| 3 $\text{GIVE}_{n=1}$ | 124.2 | 41.8 | 45.9 | 0.5 | 67 | 69 | 68 |
| 4 $\text{GIVE}_{n=2}$ | 129.9 | 83.4 | 89.9 | 0.6 | 72 | **77** | **77** |

in KG. Besides, the running time of GIVE is independent to the size of the KG, we see this if we compare its running time on small UMLS of 135 nodes and the ConceptNets which have hundreds of thousands of entities, because GIVE always selects the most important entities related to the query to induce knowledge, and the cost of the entity selection phase is very low if we pre-compute the embeddings. (2) In terms of context length, GIVE does not suffer from overwhelming long context

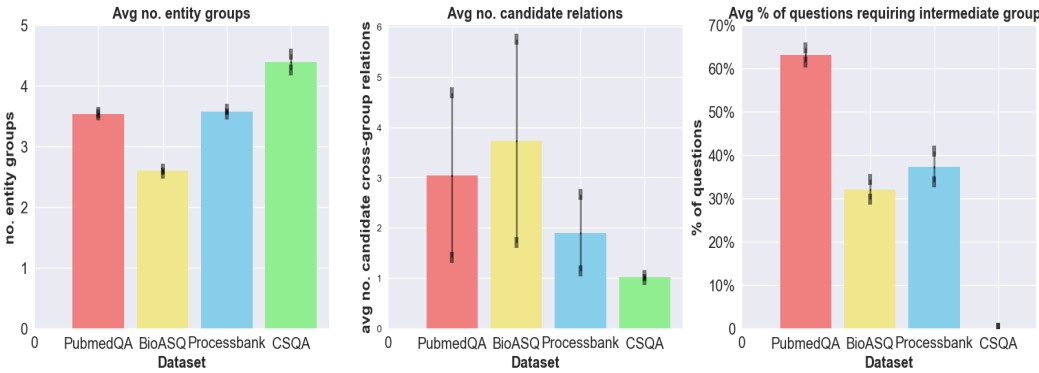

Figure 10: Average number of entity groups (left), average number of candidate relations between two groups (middle) and average percentage of questions that requires intermediate entity group (right) for each dataset included in Section 4.2 and 4.3 for 5 runs. For CSQA, we report the results on 50% triplets version of ConceptNet.

on large or dense KGs. In fact, the context length of GIVE is also largely decided by the number of relations between two groups. That is why we see that on PubmedQA where the cross-group KG knowledge is rich, GIVE can induce large number of affirmative and counterfactual knowledge, thus provides the biggest performance increase compared to RAG or ToG. It is also worth noticing the recent LLMs are making fast progress in overcoming the limitation of input length. For example, Llama 3.1 series(et al., 2024b) support up to 128k tokens, compared to Llama 3 which supports only 8,192 tokens. For GPT series models, the maximum context window size also grows from 4.1k tokens (which translates to around 3k words) of GPT3.5-turbo to 32k of GPT4 (et al., 2024a) and GPT4o. Such progress makes scaling inference time compute techniques like GIVE much more applicable, and we expect even large progress in maximum tokens on further models. GIVE is far from reaching such context length limitations according to Table 5. There are also concrete solutions to easily further reduce both running time context length of GIVE: When building the knowledge sets (Section 3.4.3), we can apply a divide-and-conquer manner to prune the knowledge in batches. When generating answers, we can apply similar techniques in GraphRAG (Edge et al., 2024), to use an additional agent to summarize the retrieved knowledge sets into shorter paragraphs before feeding to the answer generator.

**GIVE$_{n=1}$ provides a good trade-off between accuracy and efficiency.** Although the hyperparameter n=2 yields to the best accuracy in most scenarios, we see that even when we use only one additional KG entity per group, GIVE achieves better or at least the same accuracy, compared to ToG and RAG. These results further emphasize the importance of the proposed framework to incorporate structured information during inference time reasoning, at the same time, provide the practicer with a balanced alternative to use n=1 with limited compute resource, but at the same time achieve good performance.

**GIVE is able to generate high quality synthetic data using very limited external knowledge.** If we compare the accuracy increase offered by GIVE and the context length, we see a positive correlation between them. Related discussion is also included in the previous subsection that the performance of GIVE is related to the expert knowledge ratio and the number of retrieved knowledge. The results further proved that the generated knowledge is of very high-quality. As a result, GIVE has great potential to serve as a synthetic data generating algorithm in other fine-tuning tasks, such as RLHF or RL for reasoning.

### F.3 DETAILED COMPARISON WITH EXISTING RETRIEVAL METHODS

In addition to Table 1 and 2, we conduct detailed performance comparison against text-based retrieval method RAG (Lewis et al., 2021) and KG-LLM retrieval method (Sun et al., 2024), we calculate the portions of questions answered correctly by each method and present the statistics in Figure 11 and Figure 12.

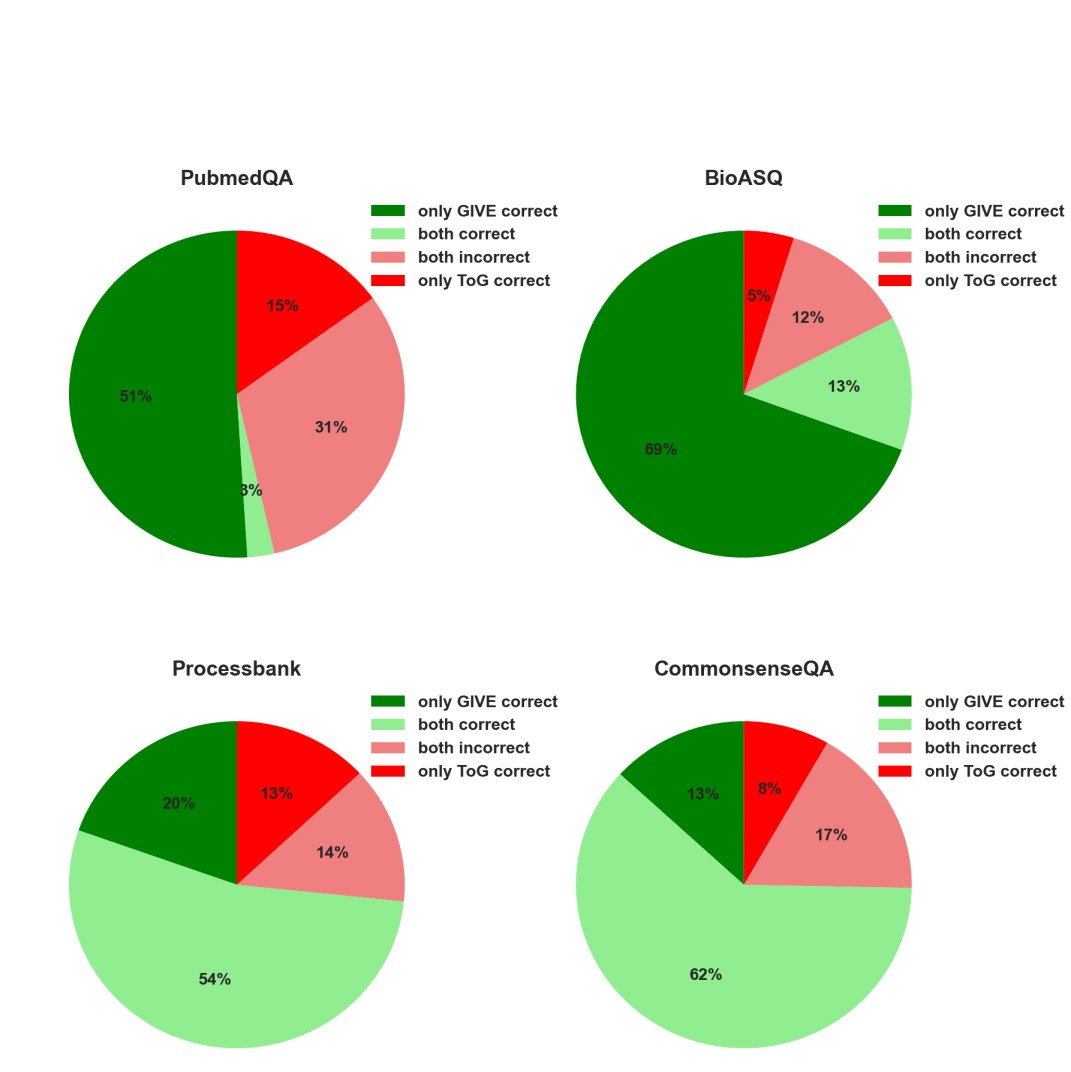

Figure 11: The proportions of questions answered correctly by GIVE and ToG, on PubmedQA, BioASQ, Processbank and CommonsenseQA

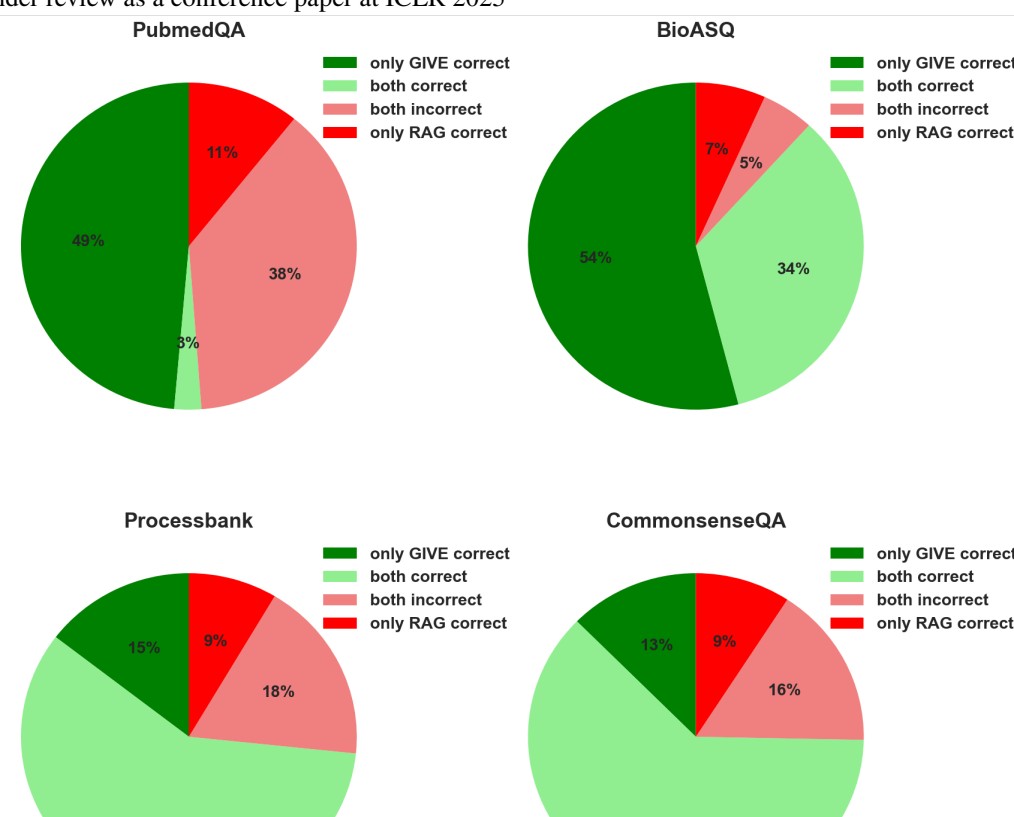

Figure 12: The proportions of questions answered correctly by GIVE and RAG, on PubmedQA, BioASQ, Processbank and CommonsenseQA

We observe that on three of the four datasets (BioASQ, Processbank, CommonsenseQA), we included in our experiments, most of the questions answered correctly by ToG or RAG is also answered correctly by GIVE. We see this by calculating the ratio $\frac{\text{only ToG/RAG correct}}{\text{only ToG/RAG correct+both correct}}$. For example, its 11% on CommonsenseQA for ToG, meaning that 89% of the questions it answered correctly is also answered correctly by GIVE. On PubmedQA, this ratio is large because RAG and ToG both get very poor performance, which means very few questions in this dataset can be directly answered by the knowledge contained in the sparse KG, this further highlights the importance of the proposed "Inspiration" process to combine internal knowledge and external knowledge to solve challenging scientific questions.

# G    PROMPTS AND EXAMPLE RESPONSES

## G.1    IO PROMPT

You are a helpful assistant that answers a given question about medical knowledge with yes, no or maybe, based on your own knowledge.
[k-shot EXAMPLES]
Q: Traumatic aortic injury: does the anatomy of the aortic arch influence aortic trauma severity?
**Output: no**

## G.2 CoT Prompt

> You are a helpful assistant that answers a given question about medical knowledge with yes, no or maybe, based on your own knowledge.
> [k-shot EXAMPLES]
> Q: Traumatic aortic injury: does the anatomy of the aortic arch influence aortic trauma severity?
> **Let's think step by step.**
> **Output: maybe**

## G.3 RAG Prompt

For RAG, we provide both the correct textual knowledge and reasoning chain for each of the k-shot examples.

> You are a helpful assistant that answers a given question about medical knowledge with yes, no or maybe, based on the retrieved textual knowledge "entity relation entity" from an expert knowledge graph.
> [k-shot EXAMPLES]
> Q: Traumatic aortic injury: does the anatomy of the aortic arch influence aortic trauma severity?
> Knowledge: [Textual knowledge]
> **Output: no**

## G.4 ToG Prompt

We follow **the official implementation of ToG (Sun et al., 2024)** and use the default prompts. We replace the k-shot examples to be examples randomly selected for each dataset, and we provide the correct reasoning chain. Overall, we use exact the same k-shot examples for ToG and our method to guarantee fair comparison.

Exemplar prompt for retrieving top entities:

> Please retrieve the top entities (separated by semicolon) that contribute to the question.
> [EXAMPLES]
> Q: Traumatic aortic injury: does the anatomy of the aortic arch influence aortic trauma severity?
> **Output: [Entities retrieved]**

Exemplar prompt for pruning relations:

> Please retrieve 1 relation that contributes to the question the most from the given relation list. The answer must be one of the given relations.
> [EXAMPLES]
> Q: Traumatic aortic injury: does the anatomy of the aortic arch influence aortic trauma severity?
> Relations: [Relations list]
> **Output: [Relationship selected]**

Exemplar prompt for pruning entities:

> Please score the entities' contribution to the question on a scale from 0 to 1 (the sum of the scores of all entities is 1). [EXAMPLES]
> Q: Traumatic aortic injury: does the anatomy of the aortic arch influence aortic trauma severity?
> Relation: [Relationship selected]
> Entities: [Entities list]
> **Output: [Entity selected]**

Exemplar prompt for evaluating knowledge sufficiency:

> Given a question and the associated retrieved knowledge graph triplets (entity, relation, entity), you are asked to answer whether it's sufficient for you to answer the question with these triplets and your knowledge (yes or no).
> [EXAMPLES]
> Q: Traumatic aortic injury: does the anatomy of the aortic arch influence aortic trauma severity?
> Knowledge triplets: [currently retrieved knowledge triplets]
> **Output: [yes/no]**

Exemplar prompt for ToG answering the question:

> Given a question and the associated retrieved knowledge graph triplets (entity, relation, entity), you are asked to answer the question with these triplets and your knowledge.
> [k-shot EXAMPLES]
> Q: Traumatic aortic injury: does the anatomy of the aortic arch influence aortic trauma severity?
> Knowledge triplets: [retrieved knowledge triplets]
> **Output: maybe**

## G.5 GRAPHRAG PROMPT

We follow **the networkx implementation of GraphRAG (Edge et al., 2024)** and use the default prompts. We replace the k-shot examples to be examples randomly selected for each dataset, and we provide the correct reasoning chain. The k-shot examples are provided during the intermediate answers generating step. Overall, we use exact the same k-shot examples for GraphRAG and our method to guarantee fair comparison.

Exemplar prompt for summarizing each detected community:

> Summarize the following community of entities and relationships.
> [Description of communities]
> **Output: [List of summaries of each community group]**

Exemplar prompt for generating intermediate answers from community summaries:

> You are a helpful assistant that answers a given biomedical question based on the provided summary.
> You can find some examples below: + [k-shot examples]
> Query: [Question]
> Summary: [Summary list]
> **Output: [List of intermediate answers]**

Exemplar prompt for combining intermediate answers into a final answer:

> You are a helpful assistant that answers a biomedical question with yes, no or maybe, based on some intermediate answers.
> Query: [Question]
> Intermediate answers: [List of intermediate answers]
> **Output: [yes/no/maybe]**

## G.6 GIVE PROMPT

Exemplar prompt for extracting and ranking the entities in the question:

> Please retrieve the top entities that contribute to the question. Answer only the top entities, separated by comma.
> [EXAMPLES]
> Question: Traumatic aortic injury: does the anatomy of the aortic arch influence aortic trauma severity?
> **Output: ['traumatic aortic injury', 'anatomy', 'aortic arch', 'aortic trauma severity']**

Exemplar prompt for extracting the relationships in the question:

> Please retrieve the relationships that connect the given entities in the question.
> [EXAMPLES]
> Question: Traumatic aortic injury: does the anatomy of the aortic arch influence aortic trauma severity?
> Entities: traumatic aortic injury, anatomy, aortic arch, aortic trauma severity
> **Output: ['influence']**

Exemplar prompt for generating relationships between two given entities:

> You are a helpful assistant that answers a short relationship in a few words between two given biomedical entities.
> [EXAMPLES]
> Entities: traumatic aortic injury, injury and poisoning
> **Output: "is a"**

Exemplar prompt for determining if relations exists between cross group entities:

> You are a helpful assistant that answers yes, no or maybe depending on the correctness of the given statement.
> Injury or poisoning is the result of organism function. Is it true?
> **Output: "No"**

Exemplar prompt for selecting optimal 2-hop path for intermediate entity group construction:

> You are a helpful assistant that selects one from the given knowledge facts (entity, relation, entity, relation, entity), that is most important to the given question.
> Knowledge Facts:
> (steroid, affects, organ or tissue function, affects, invertebrate),
> (steroid, affects, experimental model of disease, manifestation of, injury or poisoning),
> (anatomical abnormality, manifestation of, organism function, affects, clinical attribute)...
> Question to answer: Traumatic aortic injury: does the anatomy of the aortic arch influence aortic trauma severity?
> **Output: (anatomical abnormality, manifestation of, organism function, affects, clinical attribute)**

Exemplar prompt for generating GIVE$_a$:

You are a helpful assistant that answers a given question about medical knowledge with yes, no or maybe, based on the retrieved knowledge triplets (entity, relation, entity) from your own knowledge. The return must be one of yes, no or maybe.

[k-shot EXAMPLES]

Q: Traumatic aortic injury: does the anatomy of the aortic arch influence aortic trauma severity?

**[AFFIRMATIVE KNOWLEDGE TRIPLETS]**

eg: ('anatomical abnormality', 'affects', 'organism function'), ('injury or poisoning', 'affects', 'organism function'), ('anatomy', 'part of', 'aortic arch'), ('injury or poisoning', 'affects', 'organ or tissue function'), ('aortic arch', 'location of', 'injury or poisoning')...

**Output: maybe (GIVE$_a$)**

Logic Chain: I reached the answer 'maybe' by considering the relationship between the anatomy of the aortic arch and the severity of aortic trauma. The knowledge triplets suggest that the anatomy of the aortic arch may influence the severity of aortic trauma, as anatomical structure correlates with clinical attributes and impacts clinical attributes. Additionally, the severity of aortic trauma may correlate with clinical attributes, which can be affected by traumatic aortic injury. However, the relationship between the anatomy of the aortic arch and the severity of aortic trauma is not definitively stated in the knowledge triplets, hence the answer 'maybe'.

Exemplar prompt for generating GIVE$_{a+c}$:

You are a helpful assistant that answers a given question about medical knowledge with yes, no or maybe, based on the retrieved knowledge triplets (entity, relation, entity) from your own knowledge.

[k-shot EXAMPLES]

Q: Traumatic aortic injury: does the anatomy of the aortic arch influence aortic trauma severity?

**[AFFIRMATIVE KNOWLEDGE TRIPLETS]**

eg: ('anatomical abnormality', 'affects', 'organism function'), ('injury or poisoning', 'affects', 'organism function'), ('anatomy', 'part of', 'aortic arch'), ('injury or poisoning', 'affects', 'organ or tissue function'), ('aortic arch', 'location of', 'injury or poisoning')...

A: maybe (GIVE$_a$)

**Additional knowledge triplets: [COUNTER-FACTUAL KNOWLEDGE TRIPLETS]**

eg: ('organism', 'not result of', 'aortic trauma severity'), ('injury or poisoning', 'not complicates', 'anatomical structure'), ('aortic arch', 'not influence', 'injury or poisoning')...

**Output: yes (GIVE$_{a+c}$)**

Logic Chain: By utilizing the additional knowledge triplets provided, it can be inferred that the severity of aortic trauma is not a result of an experimental model of disease, steroid, traumatic aortic injury, or injury or poisoning. This suggests that the severity of aortic trauma is not influenced by these factors. Therefore, the anatomy of the aortic arch may indeed influence the severity of aortic trauma in cases of traumatic aortic injury.

Exemplar prompt for generating GIVE$_{a+c+e}$ :

---

You are a helpful assistant that answers a given question about medical knowledge with yes, no or maybe, based on the retrieved knowledge triplets (entity, relation, entity) from your own knowledge, and the knowledge triplets from an expert knowledge base. The return must be one of yes, no or maybe.

[k-shot EXAMPLES]

Q: Traumatic aortic injury: does the anatomy of the aortic arch influence aortic trauma severity?

**[AFFIRMATIVE KNOWLEDGE TRIPLETS]**

eg: ('anatomical abnormality', 'affects', 'organism function'), ('injury or poisoning', 'affects', 'organism function'), ('anatomy', 'part of', 'aortic arch'), ('injury or poisoning', 'affects', 'organ or tissue function'), ('aortic arch', 'location of', 'injury or poisoning')...

A: maybe (GIVE$_a$)

**Additional knowledge triplets: [COUNTER-FACTUAL TRIPLETS]**

eg: ('organism', 'not result of', 'aortic trauma severity'), ('injury or poisoning', 'not complicates', 'anatomical structure'), ('aortic arch', 'not influence', 'injury or poisoning')...

A: yes (GIVE$_{a+c}$)

**Additional knowledge triplets retrieved from expert knowledge base: [EXPERT KG KNOWLEDGE TRIPLETS]**

eg: ('injury or poisoning', 'result of', 'anatomical abnormality'), ('steroid', 'causes', 'injury or poisoning'), ('injury or poisoning', 'complicates', 'anatomical abnormality'), ('anatomical abnormality', 'result of', 'injury or poisoning')...

**Output: yes (GIVE$_{a+c+e}$)**

Logic Chain: I utilized the external knowledge from the expert knowledge base to modify my previous answers by considering the additional knowledge triplets that were retrieved. Aortic arch is part of anatomical structure, which is the location of injury or poisoning. Anatomical structure also affects cell function, which further affects injury or positioning, suggesting the anatomy of the aortic arch could influence the severity of aortic trauma. Therefore, the correct answer to this question should be 'yes'.

---

