# OpenReview forum: "GIVE: Structured Reasoning with Knowledge Graph Inspired Veracity Extrapolation"
_ICLR.cc/2025/Conference — Submitted to ICLR 2025_

### Official Review · Reviewer_op2q · 2024-11-02

**Soundness:** 2
**Presentation:** 2
**Contribution:** 3
**Rating:** 5
**Confidence:** 4

**Summary:**

This paper presents a novel framework called GIVE (Graph Inspired Veracity Extrapolation), aiming at improving the reasoning ability of LLMs in knowledge-intensive tasks with sparse KGs. GIVE obtains similar entities for queried entities to construct relevant entity groups, then finds inter-group and inner-group connections with LLMs. By using LLMs to extrapolate probable relationships, GIVE addresses the limitation of traditional RAG methods when reasoning on domain-specific sparse KGs. Experiments on biomedical and commonsense datasets demonstrate the effectiveness.

**Strengths:**

S1. GIVE focuses on an important and practical issue. Traditional RAG approaches show poor performance on reasoning-rich tasks when domain knowledge cannot be easily obtained and domain-specific KGs are hard to construct and maintain. The idea behind GIVE could promote the application of LLMs in specialized domains with scarce data.

S2. Massive experiments are designed to support the authors' claim. As shown in Section 4, four datasets from two different domains are involved to demonstrate the effectiveness of GIVE. From Appendix, many ablation studies and detailed analysis are performed, showing robustness. The attached prompt templates also ensure that GIVE is easy to reproduce.

S3. The organization is good. The organization of Section 3 is clear and easy to follow.

**Weaknesses:**

W1. In general, this paper is well-written. However, some parts are very brief and hard to comprehend. It is better to present more details with some equations in Section 3.3 and 3.5.

W2. Limited evaluation and incomplete analysis. While massive experiments are conducted, there are still some experiments needed to enhance robustness. As for efficiency, comparison of LLM calls is not comprehensive enough. Details are listed in Questions.

**Questions:**

Q1. Could you provide a definition of "sparse"? From Section 4.1.1, we can see that each node in UMLS connects to about 87 nodes in average. In my opinion, UMLS is a "small" KG with only 135 nodes, but it's not a sparse KG if you take the degree of nodes into account.

Q2. The authors mentioned that KBQA datasets are unsuitable because GIVE is designed for reasoning with sparse KGs. I wonder whether GIVE works for KBQA with incomplete KGs if some edges are removed, just like EmbedKGQA (https://aclanthology.org/2020.acl-main.412), one of related works mentioned in Section 2.

Q3. The authors evaluate efficiency based on the number of LLM calls. How about the context length? Could you provide the averaged number of input tokens to solve one question for GIVE.

Q4. About multi-step reasoning. I wonder why only 2-hop paths are selected? Do you experiment with longer paths?

---

> ### Author Response · Authors · 2024-11-23
>
> > In general, this paper is well-written. However, some parts are very brief and hard to comprehend. It is better to present more details with some equations in Section 3.3 and 3.5.
>
> We appreciate this advice on improving the presentation of our paper. We follow the reviewer's suggestion to add the mathematical details in the corresponding sections of our revised manuscript.
>
> > Could you provide a definition of "sparse"? From Section 4.1.1, we can see that each node in UMLS connects to about 87 nodes in average. In my opinion, UMLS is a "small" KG with only 135 nodes, but it's not a sparse KG if you take the degree of nodes into account.
>
> Thank you for raising this insightful discussion. We agree with the reviewer that UMLS is a small but dense network in the sense of graph mining. However, the "sparse" in this study refers to the fact that the precise information that directly solves the query is not included in the KG. UMLS only contains 135 nodes, so it cannot solve the biomedical queries. We see this from the failure of the retrieval-based methods, so it falls into the category of "sparse" external knowledge source. This discussion is included in the revised manuscript Section 4.1.1
>
> We are convinced that the reviewer's suggestion is valuable to make our presentation more precise. We change the description of UMLS to be "small but dense". We have three types of KGs in our experiments: (1) small and dense UMLS (2)large and dense ConceptNet (3)large and sparse ConceptNet. We further quantify the statistics of all KGs included in Appendix Section C of the revised manuscript.

---

> ### Author Response · Authors · 2024-11-23
>
> > The authors mentioned that KBQA datasets are unsuitable because GIVE is designed for reasoning with sparse KGs. I wonder whether GIVE works for KBQA with incomplete KGs if some edges are removed, just like EmbedKGQA (https://aclanthology.org/2020.acl-main.412), one of related works mentioned in Section 2.
>
> We understand the reviewer's concern about the generalizability of GIVE on KGs with missing edges.
>
> (1) The KBQA datasets focus on retrieving the information from a comprehensive knowledge base. In other words, they are designed to test the ability of the algorithm to locate the exact information related to the query. In the case of missing connections, as pointed by the authors of EmbedKGQA, the natural way to solve this problem is to utilize a link prediction algorithm to complete the KG, before using it in the retrieving pipelines. On the contrary, GIVE is not designed as a KG-completion method. In this research, we explore the possibility of combining external (RAG) and internal (reasoning) knowledge of LLMs, that is why we name the algorithm "Graph Inspired Veracity Extrapolation".
>
> (2) Although our experiments proved GIVE generalizes well in different domains (biomedical, commonsense, open-domain, see Appendix D for the new open-domain dataset TruthfulQA). We propose GIVE to tackle the reasoning-intensive questions in the hard domains where (1) obtaining enough information during training and (2) retrieving from inclusive knowledge sources are both resource-prohibited. We see this from the failure of the latest LLMs and all existing retrieval-based methods in Table 1 of Section 4.2.
>
> (3) We share the same research question with the reviewer: How does the sparsity of KG impact the performance of GIVE? We answer this question in Section 4.3 to remove edges from a full ConceptNet. The conclusion is GIVE achieves the best performance regardless of the sparsity of the KG. This is a more reasoning-intensive setting compared to the KBQA, where most questions are factual-based, so it fits more closely to the intention of this study.
>
> We modify the abstract of the paper to reflect this discussion: CoT focuses on internal knowledge reasoning, ToG relies on external resources to provide explicit logical chain, GIVE bridges these two approaches by saying: If either one is not enough, why not we combine them?

---

> ### Author Response · Authors · 2024-11-23
>
> > The authors evaluate efficiency based on the number of LLM calls. How about the context length? Could you provide the average number of input tokens to solve one question for GIVE.
>
> Thank you for raising this question, we include a detailed study of GIVE's efficiency in terms of both running time and context length in Appendix F.2 of the revised manuscript. The conclusion is (1) Running GIVE with n=1, where n is the number of KG concepts per group achieves a balanced tradeoff between efficiency and accuracy. (2) The synthetic data generated by GIVE is of high quality and contributing to the overall performance of the method. (3) The context length of GIVE is far from the limits of the mainstream LLMs.

---

> ### Author Response · Authors · 2024-11-25
> **We would like to hear your thoughts on the revised paper**
>
> Dear reviewer,
>
> Thank you for taking time to provide the constructive feedbacks. Your comments have been instrumental to help us derive new results, especially on the efficiency of GIVE,  as well as improve the presentation of our paper. We believe we have addressed all your concerns and we would extremely appreciate if you could find time to share your thoughts on our response and revised manuscript.

---

> > ### Comment · Reviewer_op2q · 2024-11-27
> >
> > Thank you for the authors' response. Most of my concerns have been solved, but the issues regarding evaluation on KBQA datasets remain unaddressed. This is a significant drawback that could limit the generalizability of GIVE, especially since ToG, one of the comparison approaches, has been evaluated on the KBQA task. The authors noted that GIVE is not designed as a KG-completion method. However, KBQA with incomplete KGs seems to be an ideal application for GIVE. LLMs are able to find some missing facts through combining the internal knowledge of LLMs and the external knowledge of KGs. The combination of knowledge from LLMs and KGs is claimed as the main contribution of GIVE. I sincerely recommend that the authors include experiments on KBQA tasks to provide a more comprehensive evaluation.

---

> > > ### Author Response · Authors · 2024-11-27
> > >
> > > > especially since ToG, one of the comparison approaches, has been evaluated on the KBQA task
> > >
> > > ToG, as explained by the authors of the paper, is an iterative KG explorer algorithm that reasons based on the retrieved knowledge. In other words, it is designed to solve question (1).  It also has great and well-proved generalizability, and its generalizability does not contradict with GIVE’s, because they are solving different problems.
> > >
> > > We include ToG as baseline, not to criticize its generalizability, but to show that there exists real-world settings that it is not designed for, that’s what motivates us to propose methods on different axis, like GIVE.

---

> > > ### Author Response · Authors · 2024-11-27
> > >
> > > >  The authors noted that GIVE is not designed as a KG-completion method. However, KBQA with incomplete KGs seems to be an ideal application for GIVE.
> > >
> > > We are glad that the reviewer see great potential of GIVE. We agree with the reviewer that methods like GIVE could be a fit for different applications wherever there is the issue of shortage of data. These problems include incomplete KBQA, synthetic data generation, etc, as already been discussed in our manuscript. However, such potential of GIVE in all these different research questions is one of the advantages of our method, not drawbacks. And that is why we feel excited about our research to be able to inspire further investigation in this domain. It is unfair for GIVE to be excluded from the community because not all   problems it inspires are empirically tested in one single paper, given the intensive experiments that have been conducted, regarding the foundation of its ability to solve these potential further problems.

---

> > > ### Author Response · Authors · 2024-11-27
> > >
> > > > LLMs are able to find some missing facts through combining the internal knowledge of LLMs and the external knowledge of KGs.
> > >
> > > We agree with this insightful discussion raised by the reviewer, and we believe using LLMs for KG-completion is a promising direction, but this is beyond the discussion of this study. We pointed out that “GIVE sheds light on the potential of LLM to conduct divergent reasoning using very limited external clues.” We hope to inspire further research in the relevant domains, that’s why we think GIVE should be exposed to the community for discussion.

---

> > > ### Author Response · Authors · 2024-11-27
> > >
> > > > The combination of knowledge from LLMs and KGs is claimed as the main contribution of GIVE.
> > >
> > > We raise this argument because reasoning purely on internal/external knowledge is not enough. Based on this observation, GIVE is proposed to solve problem (2). We believe there will be further research that solve KG-completion or other related problems based on the same observation. Again, that’s why we are eager to raise this to the community.

---

> > > ### Author Response · Authors · 2024-11-27
> > >
> > > > I sincerely recommend that the authors include experiments on KBQA tasks to provide a more comprehensive evaluation.
> > >
> > > We sincerely appreciate the reviewer’s dedication in providing all the constructive feedback for our paper and we sincerely understand that the reviewer’s comments are for the good of this study.
> > > As discussed, the principle raised by GIVE is “we need to combine training and inference time knowledge”. This principle can be applied to a few potential applications. KBQA, as an instance of (1), is fundamentally different from the problems that we have been solving defined by (2) and we hope the reviewer understand that it is a major new experiment. At this moment, we are prioritizing engagement in the discussion, which as we said, has been fruitful and enjoyable.

---

> ### Author Response · Authors · 2024-11-27
>
> Dear reviewer,
>
> Thank you for your continuous engagement in the discussion and we are glad that most of your concerns have been solved by our rebuttal. At this point, the reviewer and us are in agreement that the method we propose is valuable. We understand the reviewer’s thinks about the generalizability of GIVE and we would like to further address this concern in the following perspectives:
>
> > This is a significant drawback that could limit the generalizability of GIVE
>
> We share with the reviewer that there are different tasks that require integration of LLM knowledge and external knowledge, two of which are significant: (1) factual-rich tasks when inclusive KB is available (2) reasoning-rich tasks when domain knowledge cannot be obtained. This is pointed out by S1 in the original review as well.
>
> With, GIVE is designed to solve question (2), and “generalizability” is defined with respect to the research question a study is designed to solve [1] [2].  Although we design GIVE with sparse KG and scientific domain tasks in mind (because we want to solve the generally recognized problem (2) that has been ignored, which is also agreed by all reviewers), we validate GIVE on QA tasks from different domains, on KGs of different sizes and sparcities. As suggested by the reviewer, we further conduct detailed validation of efficiency, even if the word “reasoning” itself implies longer time, and richer context (this holds for human intelligence, so it should hold for AI).
>
> All these studies have proved GIVE’s great generalizability in terms of both accuracy and efficiency. With all due respect, we are convinced that this evidence serves as a fair judgement that GIVE is generalizable.
>
> [1] Polit DF, Beck CT. Generalization in quantitative and qualitative research: myths and strategies.
>
> [2] Understanding Generalizability and Transferability - The WAC Clearinghouse.

---

### Official Review · Reviewer_7idj · 2024-11-04

**Soundness:** 3
**Presentation:** 3
**Contribution:** 3
**Rating:** 8
**Confidence:** 3

**Summary:**

The paper introduces GIVE (Graph Inspired Veracity Extrapolation), a structured reasoning framework aimed at enhancing the factual reliability of large language models (LLMs) when working with sparse knowledge graphs (KGs). Recognizing the limitations of LLMs in specialized or scientific domains where comprehensive KGs are infeasible, GIVE leverages both internal and external knowledge sources. It decomposes queries into core concepts, creates entity groups for each query, and induces reasoning chains by identifying potential relationships among these groups. The framework incorporates both factual and counterfactual connections to mitigate hallucination risks in reasoning. Experiments on biomedical and commonsense QA benchmarks demonstrate that GIVE enables LLMs, such as GPT-3.5-turbo, to surpass even more advanced models like GPT-4 without additional training costs

**Strengths:**

The paper is clear and well-organized, making the technical contributions easy to follow.

Given the increasing reliance on LLMs for knowledge-intensive tasks, addressing the factuality of LLM outputs in specialized domains is a critical area of research.

The GIVE framework introduces an innovative approach by combining structured KG-inspired reasoning with parametric and non-parametric memories, presenting an original solution to a well-recognized problem in LLM performance.

**Weaknesses:**

The framework appears quite complex, combining multiple distinct modules into a single system. This layered approach raises concerns about the simplicity and elegance of the solution, as well as its general applicability.

 Although the paper includes a few ablations, the complexity of the approach warrants a more thorough exploration of each module's contribution to overall performance. For instance, it would be valuable to have a clearer understanding of how each step (e.g., inner-group and inter-group connections, counterfactual relations) affects the final results.

**Questions:**

How does GIVE perform in comparison to GraphRAG or similar frameworks that incorporate KG information into LLM-based reasoning? Since GraphRAG is specifically designed for KG integration, this would be a valuable comparison to include.

Why do some baselines without external knowledge outperform those with retrieval? It raises the question of whether the implementation of the baselines was correct. Could the authors clarify if there might be any explanation for these results?

GIVE is tested on specialized datasets, but commonly used benchmarks such as TruthfulQA and Natural Questions (NQ) are not included. Testing GIVE on these datasets would provide a broader perspective on its generalizability and effectiveness in widely used tasks.

The paper compares GIVE primarily against RAG and ToG. However, there is a substantial amount of literature on KG-enhanced LLMs. Adding comparisons with other KG-based enhancements would strengthen the experimental results and help position GIVE more accurately within the state of the art.

---

> ### Author Response · Authors · 2024-11-23
>
> >The framework appears quite complex, combining multiple distinct modules into a single system. This layered approach raises concerns about the simplicity and elegance of the solution, as well as its general applicability.
>
> We understand the reviewer's concern about the "general applicability" of our method. We include KGs of different sizes and sparcities (small but dense UMLS, large dense ConceptNet, large sparse ConceptNet) to solve questions in different domains (biomedical, commonsense, open-domain) of different types (yes-no, multiple choice, text generation). In every one of these comparisons, our method achieves the best performance compared to the recent baselines, which proved the general applicability of our approach.
>
> > it would be valuable to have a clearer understanding of how each step (e.g., inner-group and inter-group connections, counterfactual relations) affects the results.
>
> We appreciate the reviewer's suggestion to enhance our ablation studies. We include new ablations in Appendix E.1 of our revised manuscript on GIVE using (1) only inner group connections (2) only inter group connections (3) both inner and inter group connections. The conclusion is they are both important for GIVE to achieve the best performance, whereas the inter group connections contribute more to the overall success of GIVE because they are the knowledge source to reduce the faithful multi-hop reasoning chain, whereas the inner group connections serve as an intermediate to bridge the query and the sparse KG.

---

> ### Author Response · Authors · 2024-11-23
>
> > How does GIVE perform in comparison to GraphRAG or similar frameworks that incorporate KG information into LLM-based reasoning? Since GraphRAG is specifically designed for KG integration, this would be a valuable comparison to include.
>
> We thank the reviewer's suggestion. GraphRAG is a RAG technique to retrieve knowledge on the information network by bridging the connections between each individual information unit. GIVE aims to prompt faithful reasoning in hard domains where training and retrieving on inclusive resource is prohibited.
>
> However, we agree with the reviewer that comparing with GraphRAG will be valuable to showcase the effectiveness of our method. We include this comparison in Table 1 of Section 4.2. Since GraphRAG is proposed to operate on the information network, we exclude it from the large-scale experiments utilizing ConceptNet.
>
> > Why do some baselines without external knowledge outperform those with retrieval? It raises the question of whether the implementation of the baselines was correct. Could the authors clarify if there might be any explanation for these results?
>
> We appreciate this insightful discussion raised by the reviewer. The retrieval methods fail heavily because they cannot retrieve useful information from the sparse knowledge graph. We see this from the examples given in Figure 1 of our revised manuscript, thus causes hallucination. GIVE resolves this issue by first bridging the query and KG using inner-group connections and then use the KG knowledge to populate the inter-group connections between the retrieved entity groups.

---

> ### Author Response · Authors · 2024-11-23
>
> > GIVE is tested on specialized datasets, but commonly used benchmarks such as TruthfulQA and Natural Questions (NQ) are not included. Testing GIVE on these datasets would provide a broader perspective on its generalizability and effectiveness in widely used tasks.
>
> We thank the reviewer for their suggestion. The principle of our experiments is to use the KG that is related to the domain of the questions but does not provide direct answer to solve the queries, because this is the most realistic scenario for human/expert problem solving, to conduct deductive reasoning using very limited external hints. That is why we include these biomedical reasoning tasks where obtaining enough information during training is extremely hard, and building inclusive corpus for inference time retrieval is also infeasible.
>
> Following the reviewer's suggestion, we conduct additional experiments on TruthfulQA using 10\% edge ratio ConceptNet, and the results are presented in Appendix D of the revised manuscript. We conclude that GIVE uniformly achieves the best performance regardless of the domain/format of the dataset and size/sparsity of the KG.
>
> > The paper compares GIVE primarily against RAG and ToG. However, there is a substantial amount of literature on KG-enhanced LLMs. Adding comparisons with other KG-based enhancements would strengthen the experimental results and help position GIVE more accurately within the state of the art.
>
> We understand the reviewer's concern about our baselines. As discussed in Section 2, existing methods heavily focus on the task of KBQA, where the task is to identify the direct information that can solve the query. That is why they are built on the assumption that the given external source is inclusive, so they are not designed for the tasks we are trying to solve. We compare against ToG and GraphRAG, which is the latest best-performing methods for reasoning on KG and retrieving on structured data.

---

> ### Author Response · Authors · 2024-11-25
> **We would like to hear your thoughts on our response and revised manuscript**
>
> Dear reviewer,
>
> We would like to express our gratitude for your appreciation of our work. Your comments were to the point and we used them to conduct a number of additional experiments to improve our study. We would really appreciate if you could share your thoughts with us.

---

> ### Comment · Reviewer_7idj · 2024-11-26
>
> Thanks for the clarification and the revision. It seems that my concerns are properly addressed. I will update my score accordingly.

---

> > ### Author Response · Authors · 2024-11-26
> > **Thank you for your review and engagement**
> >
> > Dear reviewer,
> >
> > Thank you for taking time to review our paper and participate in the discussion. We appreciate your acknowledgement of our work. Your feedback and inputs have been valuable for us to improve the paper and we are glad to answer your questions.

---

### Official Review · Reviewer_Z1Eq · 2024-11-04

**Soundness:** 3
**Presentation:** 3
**Contribution:** 2
**Rating:** 6
**Confidence:** 3

**Summary:**

This paper introduces GIVE, a novel framework that enhances LLM's reasoning capabilities by leveraging sparse knowledge graphs. Rather than relying on direct knowledge retrieval, GIVE uses KG structure as "inspiration" for LLMs to conduct structured reasoning. The framework operates by decomposing queries into key concepts, constructing entity groups, and building connections within and across these groups through a combination of KG guidance and LLM knowledge.
The paper's main contribution is a new paradigm that bridges parametric (LLM internal knowledge) and non-parametric (external KG) memories for structured reasoning, while preventing hallucination through counterfactual knowledge incorporation. GIVE employs a progressive answer generation approach that combines affirmative, counter-factual, and expert knowledge to produce reliable responses. Through extensive experiments on both biomedical and commonsense reasoning tasks, the authors demonstrate that GIVE enables GPT3.5-turbo to outperform more advanced models like GPT4 on biomedical reasoning using only a sparse knowledge graph, without additional training.

**Strengths:**

The paper exhibits several notable strengths in its motivation, methodology, and empirical findings.

**Clear Motivation**: The authors identify a critical gap in current retrieval-based reasoning approaches for LLMs, which typically require dense, high-quality knowledge sources. This requirement poses a significant challenge in specialized domains like healthcare where comprehensive knowledge bases are expensive or impractical to build. The paper's focus on enabling effective reasoning with sparse knowledge graphs addresses a real-world constraint in deploying LLMs for domain-specific applications.

**Novel and Intuitive Method**: The proposed GIVE framework presents an elegant solution by treating knowledge graphs as inspiration rather than mere information sources. The method intuitively mimics human expert reasoning by first identifying relevant concepts, building semantic groups, and then discovering relationships both within and across these groups. The progressive answer generation strategy, which incorporates affirmative, counter-factual, and expert knowledge, provides a natural way to prevent hallucination while leveraging both the LLM's internal knowledge and external information.

**Interesting Findings**: The experimental results reveal several compelling insights. Most notably, GIVE enables GPT3.5-turbo to outperform GPT4 on biomedical reasoning tasks using only a sparse knowledge graph of 135 nodes, demonstrating that structured reasoning can compensate for model size. The ablation studies also reveal an interesting correlation between performance and expert knowledge ratio, suggesting that even a small amount of expert knowledge can effectively guide LLM reasoning when properly structured.

**Weaknesses:**

**Limited Scope and Comprehensive Evaluation**: While GIVE is presented as a general framework applicable across domains, the empirical evaluation heavily focuses on medical QA tasks with sparse KGs. The experiments on CommonsenseQA with denser KGs are limited to GPT-3.5-turbo and lack comprehensive comparisons with competitive baselines. Despite claiming effectiveness "in retrieving information from both sparse and dense KG," there isn't sufficient experimental evidence demonstrating how GIVE helps bridge the performance gap between smaller and more advanced LLMs across different KG densities. A more thorough evaluation across different model sizes and KG configurations would better support the framework's generalizability claims.

**Scalability and Computational Analysis**: The paper's experiments primarily focus on sparse KGs with relatively small numbers of entity groups (m) and entities per group (n), typically less than 5 for biomedical datasets. While the authors mention potential optimizations through batch pruning, the paper lacks a thorough analysis of how the method scales with increasing m and n when applied to larger or denser KGs. The computational complexity of O(rm²n²) could become significant with larger KGs, and the paper would benefit from a detailed cost comparison with existing methods like RAG and ToG. This analysis is crucial for understanding the practical applicability of GIVE in real-world scenarios with larger knowledge bases.

**Knowledge Conflict Resolution**: The paper does not adequately address how GIVE handles conflicts between LLM's internal knowledge and KG-sourced information. While the progressive answer generation approach incorporates both knowledge sources, there isn't a clear mechanism or analysis of how the system resolves contradictions when they arise. This becomes particularly important in specialized domains where LLM's pre-trained knowledge might conflict with domain-specific KG information, potentially affecting the reliability of the reasoning process. A systematic approach to knowledge conflict resolution would strengthen the framework's robustness.

**Questions:**

1. **Comprehensive Evaluation and Generalization**
   - Could you provide additional experimental results comparing GIVE's performance across different LLM sizes (e.g., GPT-4, GPT4o-mini, Llama) on CommonsenseQA with the full ConceptNet?
   - What is the performance gap between smaller and larger LLMs when using GIVE with denser KGs? This would help substantiate the claim about GIVE's effectiveness across different KG densities.

2. **Scalability and Efficiency**
   - Could you provide an analysis of how computational costs scale when applying GIVE to denser KGs? Specifically, how does GIVE's running cost compared to RAG and ToG in the current setting and in a denser KG setting?
   - What is the practical upper limit for m and n on a large KG where GIVE remains competitive against RAG (probably with pruning as mentioned)?

3. **Knowledge Conflict Resolution**
   - How does GIVE handle cases where the LLM's internal knowledge directly contradicts the KG information? Is there a way to evaluate the reliability of conflicting information from different sources?
   - Could you provide examples of how the progressive answer generation approach handles such conflicts in practice?

---

> ### Author Response · Authors · 2024-11-23
>
> > While GIVE is presented as a general framework applicable across domains, the empirical evaluation heavily focuses on medical QA tasks with sparse KGs.
>
> >Despite claiming effectiveness "in retrieving information from both sparse and dense KG," there isn't sufficient experimental evidence demonstrating how GIVE helps bridge the performance gap between smaller and more advanced LLMs across different KG densities
>
> > What is the performance gap between smaller and larger LLMs when using GIVE with denser KGs? This would help substantiate the claim about GIVE's effectiveness across different KG densities.
>
> We understand the reviewer's concern about GIVE's effectiveness "in retrieving information from both sparse and dense KG", and "across different KG densities". We answer this question by including 3 kinds of KGs: (1) small and dense, the UMLS KG used in Section 4.2. (2) large and dense, the full ConceptNet used in Section 4.3 (3) large and sparse, the 50\% and 10\% ConceptNet in Section 4.3 and Appendix D. To further backup this claim, we include QA datasets of different types: Yes-No(PubmedQA, BioASQ), MC(Processbank, CSQA), Text generation(TruthfulQA). In every one of these comparisons, GIVE achieves the best performance regardless of the size/sparcities of the KG, the domain of the question, or the format of the specific dataset.
>
> The reason we include different LLMs in Section 4.2 is as we increase the model size, the knowledge level in scientific domains will be very hard to improve, compared to open domains like commonsense or truthfulness, because there are few available resources on the internet, whereas it is relatively easier to find an inclusive commonsense document. We see this in Table 1 if we compare the performance of GPT3.5, GPT4 and GPT4o on PubMedQA. That is also the main motivation for us to propose GIVE, which is to solve scientific domain questions where the inclusive knowledge base is hard to build, both for training and retrieving. GIVE is not proposed to bridge all the parametric gaps between small and large LLMs, but we are convinced the facts that (1) GIVE enables LLMs of different sizes to achieve better performance (2) GIVE enables smaller LLM to outperform larger ones in these hard domains where obtaining information during training is difficult are extremely encouraging. These findings provide another axis for the scaling law of LLM: if training parametric and retrieving non-parametric knowledge are both hard, why not we combine both? On top of that, Section 4.3 is included to showcase the robustness of GIVE in handling KGs of different sizes and sparcities. And another open-domain QA dataset in added in Appendix D. We also include this discussion in our revised manuscript.

---

> ### Author Response · Authors · 2024-11-23
>
> > Could you provide an analysis of how computational costs scale when applying GIVE to denser KGs? Specifically, how does GIVE's running cost compare to RAG and ToG in the current setting and in a denser KG setting?
>
> >What is the practical upper limit for m and n on a large KG where GIVE remains competitive against RAG (probably with pruning as mentioned)?
>
> We appreciate the reviewer's suggestion to include more analysis on the efficiency of GIVE. We include a comprehensive study in Appendix F.2, to compare GIVE's efficiency with RAG and ToG in all settings. The efficiency of GIVE remains reasonable as we increase the size or density of the KG. If we use n=1, GIVE achieves a balanced tradeoff between efficiency and accuracy, in all datasets. We also proposed easy methods to further improve GIVE's efficiency, through divide-and-conquer pruning knowledge, or summarization before answer generation. Also note that m is dataset-dependent, and we also proved that it is also upper-bounded by a small number in Appendix F.2. And when we increase the density of the KG, the only factor that will change is "r", which is the number of distinct connections between two groups, the increase of which is strictly upper bounded by the increase of total number of KG edges. We refer to these discussions in Appendix F.2 of our revised manuscript for the reviewer's reference.

---

> ### Author Response · Authors · 2024-11-23
>
> > How does GIVE handle cases where the LLM's internal knowledge directly contradicts the KG information? Is there a way to evaluate the reliability of conflicting information from different sources?
>
> We appreciate this insightful discussion raised by the reviewer.
>
> (1) There are very few (if any) cases the LLM directly conflicts with KG, because the ground truth KG connection is not pruned by the LLM, but directly appended to the expert knowledge set. And if there is the KG connection (u,r,v) between two entities, when we ask LLM to fill the relationship between u and v, it is very unlikely that the model will answer something that is opposite to "r", in most cases, it will answer some synonyms of r.
>
> (2) We encourage the LLM to answer "unrelated" for "discovering open relations", and "maybe" in the "veracity extrapolation" process, by saying it explicitly in the prompt and providing examples, to handle the case that it does not have enough parametric knowledge to induce this information.
>
> (3) The progressive answer generation process provides the KG knowledge in the last and tell the model that they are retrieved from an "expert knowledge base". This enables the answer generator to have a chance to correct the previous answer by providing the ground truth information in the last. According to our experiments, ${GIVE}_{a+c+e}$ achieves the best performance in most cases, which proved the effectiveness of this approach in handling knowledge confliction and providing high-level supervision, as pointed out by the reviewer.

---

> ### Author Response · Authors · 2024-11-23
>
> > Could you provide examples of how the progressive answer generation approach handles such conflicts in practice?
>
> Yes, the example that is included in Appendix G.5 and Figure 2 is an example of the case where the reduced affirmative and counter-factual knowledge are not sufficient to result in a correct answer or reasoning chain. In this case, the KG knowledge serves as a high-level hint to bridge this gap. We further append the reasoning chain of each step for the reviewer’s reference.

---

> ### Author Response · Authors · 2024-11-25
> **We would like to hear your thoughts on the revised paper**
>
> Dear reviewer,
>
> Thank you for taking time to review our paper and provide the constructive feedbacks. Our discussions have been fruitful for us and we believe we have addressed all your concerns. We would extremely appreciate if you could find time to share your thoughts on our response and the revised manuscript.

---

> > ### Comment · Reviewer_Z1Eq · 2024-11-26
> > **Thanks for the response**
> >
> > I appreciate the authors for their responses and additional experiments, which address most of my concerns. I will raise my score to 6.

---

> > > ### Author Response · Authors · 2024-11-26
> > > **Thank you for your engagement**
> > >
> > > Thank you for taking time review our paper and engage in the discussion. Your inputs have been valuable and we are glad to answer your concerns.

---

### Official Review · Reviewer_qXMg · 2024-11-11

**Soundness:** 2
**Presentation:** 2
**Contribution:** 3
**Rating:** 5
**Confidence:** 4

**Summary:**

This work aims to enhance RAG by systematically integrating external knowledge from KGs and leveraging the model's internal knowledge. It induces potential relations between different entity groups, identifies intermediate nodes for multi-hop reasoning, and employs a guided reasoning process that combines affirmed knowledge, counters potential inaccuracies, and incorporates ground-truth data.

**Strengths:**

1. The authors provide a clear motivation and a good introduction to the problem.
2. Using LLMs to generate divergent thinking on additional entities and relationships to mitigate the sparsity problem in KG-RAG is a valuable approach worth exploring.
3. The performance of the method is overall positive.

**Weaknesses:**

1. Limited novelty, as leveraging triple-based reasoning in RAG and iterative self-checking are already common practices.
2. There are relatively few evaluation datasets and baselines. (See Questions)

**Questions:**

1. Using LLMs for selection will inevitably be constrained by the input length. Considering the case where there are a large number of 2-hop path combinations between two nodes, this method seems only suitable for very sparse knowledge graphs (where there are few 2-hop path combinations on average).
2. There are relatively few evaluation datasets, and the experiments are based on scenarios involving incomplete domain-specific knowledge graphs. It remains to be seen whether it can also be competitive against baselines in open-domain benchmarks with larger KG.
3. ToG assumes that it is based on a relatively large and complete knowledge graph. Considering the issue of corpus graph sparsity, could it be compared with methods like GraphRAG(From Local to Global: A Graph RAG Approach to Query-Focused Summarization)?
4. In addition to Questions_2, it's necessary to quantify the sparsity of the knowledge graphs mentioned in this paper.

---

> ### Author Response · Authors · 2024-11-23
>
> > Limited novelty, as leveraging triple-based reasoning in RAG and iterative self-checking are already common practices.
>
> Thank you for raising this valuable discussion. We understand the reviewer's concern about the novelty of our work, and we agree that there has been works leveraging triplets or structured information to enhance RAG and information retrieval, as discussed in Section 2 and Appendix B of our revised manuscript. In this paper however, we provide a solution to the unsolved problem: How to boost the reasoning ability of LLMs using very limited external clue? GIVE is proposed as an inference time computing method to boost reasoning, not as a RAG framework for information retrieval.
>
> GIVE is not proposed to leverage KG triplets in RAG. To the best of our knowledge, this is the first attempt to use retrieved information not as golden context or long answer, but as high-level evidence to guide LLM to utilize its parametric knowledge. As a result, we provide an elegant solution to retrieve and reason on very sparse KGs, whereas all previous methods fail heavily, because there is no linkage between the query and the sparse knowledge retrieved. Besides, GIVE also sheds light to the potential of combining parametric knowledge and non-parametric evidence to generate high-quality synthetic data.

---

> ### Author Response · Authors · 2024-11-23
>
> > Using LLMs for selection will inevitably be constrained by the input length. Considering the case where there are a large number of 2-hop path combinations between two nodes, this method seems only suitable for very sparse knowledge graphs (where there are few 2-hop path combinations on average).
>
> We appreciate this insightful discussion raised by the reviewer and we agree that LLMs are restricted by the context length.
>
> In Section 4.2, we proved GIVE is effective on reasoning on small KGs, whereas in Section 4.3, we test GIVE on ConceptNet with different sparcities (edge ratios). Even though these results proved GIVE can handle both sparse and dense KG, we design our method with sparse KG in mind, the reason is we see efficiently utilizing external resources as an important problem when we scale inference time compute, as pointed out by the reviewer that "mitigate the sparsity problem in KG-RAG is a valuable approach worth exploring".
>
> In all these experimental settings, we never exceed the maximum context length. We provide the detailed context length comparison between GIVE and other method in Appendix Section D.2 of the revised manuscript. However, we agree with the reviewer that such problem may happen if we provide GIVE with a KG where "there are a large number of 2-hop path combinations". In this case, we can always prune the 2-hop paths in a divide-and-conquer manner, which means we select the most important 2-hop paths in batches, and we further select the most important one out of the previous selected ones. We also included these discussions about the possible ways to reduce GIVE's context length in the limitation statement of our revised manuscript.
>
> It is also worth noticing that recent LLMs are making fast progress in terms of maximum context window size. For example, Llama 3.1 series have a context length of 128k tokens. The context window grows from 4.1k (around 3k words) for GPT3.5-turbo to 32k for GPT4. The substantial increase from previous models make scaling inference time compute techniques such as GIVE applicable, and we can reasonably expect an even large increase in the context length for further LLMs. According to our new experiments in Appendix F.2, GIVE is not even close to these limits. We also provide possible easy solutions to reduce the context length of GIVE, which is to include summarization agents before answer generation.

---

> ### Author Response · Authors · 2024-11-23
>
> > There are relatively few evaluation datasets, and the experiments are based on scenarios involving incomplete domain-specific knowledge graphs. It remains to be seen whether it can also be competitive against baselines in open-domain benchmarks with larger KG.
>
> We thank the reviewer for raising this concern about our experiments. The principle of our experiments is to use the KG that is related to the domain of the questions, but does not provide direct answer to solve the queries, because this is the most realistic scenario for human/expert problem solving, to conduct deductive reasoning using very limited external hints. We include this discussion in the revised manuscript.
>
> We agree with the reviewer that including more open-domain datasets will enhance our evaluation. We test GIVE on another open-domain benchmark TruthfulQA, a sentence generation task QA dataset covering domains including health, law, conspiracies, and fiction. We use the 10\% ConceptNet to run GIVE and use GPT4o to compare the winning rates. This experiment is included in Appendix D of the revised manuscript, and the conclusion is GIVE also outperforms the baseline methods by a great margin in this task. Thus, we have an inclusive set of datasets which consist of biomedical, commonsense, and open-domain, and the task type spans from yes-no, multiple choice and text generation.
>
> >  Considering the issue of corpus graph sparsity, could it be compared with methods like GraphRAG(From Local to Global: A Graph RAG Approach to Query-Focused Summarization)?
>
> We appreciate the reviewer's suggestion to enhance our experiments. GraphRAG is a RAG method that retrieves information from documents, where the nodes in the graph are units of information. On the other hand, GIVE locates the most relevant information in the KG and use that to guide reasoning. We include this discussion in Appendix B of the revised manuscript.
>
> Although GIVE and GraphRAG are designed to solve different problems, we agree with the reviewer that adding this comparison is valuable to prove the effectiveness of our method, we include GraphRAG as an additional Graph-based retrieval baseline in Table 1 of Section 4.2 of the revised manuscript. Note that because GraphRAG is designed to handle document/information unit graphs, we exclude from the large-scale experiments in Section 4.3 where we use ConceptNet of more than 844k nodes.
>
> > In addition to Questions_2, it's necessary to quantify the sparsity of the knowledge graphs mentioned in this paper.
>
> We thank the reviewer for their suggestion. We have dataset statistics in Section 4.1.1 of our original manuscript. To make our presentation clearer, we present detailed statistics of the knowledge graphs used in our experiments in Appendix Section C of the revised manuscript.

---

> ### Author Response · Authors · 2024-11-25
> **We would like to hear your thoughts on our response and revised manuscript**
>
> Dear reviewer,
>
> We would like to thank you for your  constructive feedback, which helps us derive new results and enhance the quality of our paper, especially on new baseline results and efficiency of GIVE.
>
> We believe that we have addressed all of your concerns and we would really appreciate if you could find time to share your thoughts and reconsider our response and revised manuscript.

---

> ### Comment · Reviewer_qXMg · 2024-11-28
>
> Thank you for your detailed response. I have a few additional suggestions and questions for further clarification:
>
> * Quality and Completeness of Triple Chains: Given the limited availability of external resources, I am particularly interested in the exploration of the quality and completeness of the triple chains used. Could the authors systematically analyze the ability of LLMs to complement incomplete knowledge graphs? For instance, it would be valuable to understand the ratio of generated (inferred) entities/relations to retrieved entities/relations, as well as the quality and success rate of these completions under different levels of KG completeness.
>
> * Additionally, it would be beneficial to evaluate how well the proposed approach performs on a knowledge graph completion (KGC) task. This could provide more insight into the model's capability to infer missing entities and relations.
>
> * Analysis of Bad Case Patterns: It would be helpful if the authors could categorize and present the observed bad case patterns.
>
> * Reproducibility is crucial. I strongly recommend that the authors release an anonymized code repository to enable the community to validate the findings. Additionally, please provide detailed guidance including configurations for both your method and the baselines.

---

> ### Author Response · Authors · 2024-11-28
>
> We are glad to answer your concerns, and we believe the reviewer and us agree that GIVE is valuable to solve the research question: How do we incorporate limited external knowledge to guide LLM reasoning? We understand the reviewer’s further questions and we would like to provide clarification:
>
> > I am particularly interested in the exploration of the quality and completeness of the triple chains used.
>
> From Table 5 in the appendix, we observe that the amount of performance gain offered by GIVE is positively related to the context length, which proves the great quality of the synthetic data it generates (otherwise they will cause hallucination). Similar observations are also drawn in Appendix F.1, we refer to the corresponding sections of the paper for the reviewer’s reference.
>
> >Could the authors systematically analyze the ability of LLMs to complement incomplete knowledge graphs?
>
> >Additionally, it would be beneficial to evaluate how well the proposed approach performs on a knowledge graph completion (KGC) task. This could provide more insight into the model's capability to infer missing entities and relations.
>
> Use LLM for KG-completion is a promising research question but is beyond the discussion of this paper, in which we study how to use limited external resources to guide LLM reasoning. We never intend to complete the knowledge graph. Besides, with all due respect, we hope the reviewer understand that at this point, this is a new major experiment by the end of the paper-revision period, and we are prioritizing engagement in the discussion.
> The fact that GIVE provides insights to other related applications is a strength of the proposed method, not a drawback. We have included extensive experiments to showcase the effectiveness regarding the proposed research problem and we believe that we have resolved all concerns regarding this. It is unfair to exclude GIVE from discussion if not all further research questions it inspires are answered in one single paper.
>
> >For instance, it would be valuable to understand the ratio of generated (inferred) entities/relations to retrieved entities/relations
>
> Please refer to Apppendix F.1 for a detailed study of the ratio of generated knowledge to expert knowledge. This study is included in our manuscript. Again, we conduct these experiments to provide a detailed analysis on the factors that influence GIVE’s performance, not to study KG-completion.
>
>
> >as well as the quality and success rate of these completions under different levels of KG completeness.
>
> KG-completion, as a separate problem, has been well studied, and is not related to the research question raised in our study. We agree with the reviewer that it is crucial to study GIVE’s performance under different levels of KG completeness, so we conducted experiments in Section 4.3 to provide a detailed analysis.
>
> >Analysis of Bad Case Patterns: It would be helpful if the authors could categorize and present the observed bad case patterns.
>
> We agree that it is important to study in which case the performance gain offered by GIVE is less significant. We append this study in Appendix F.1. GIVE’s performance is positively related to the expert ratio. The reason is the more KG knowledge retrieved, the more high-quality synesthetic knowledge we can generate (because there are more “hints” regarding the entities and relations), to solve the query. Similar observations are also drawn from Table 5 in the appendix.
>
> >Reproducibility is crucial. I strongly recommend that the authors release an anonymized code repository to enable the community to validate the findings. Additionally, please provide detailed guidance including configurations for both your method and the baselines.
>
> We agree with the reviewer that open source in research is crucial. We include detailed configurations and prompts for baselines and GIVE in both Section 4.1.2 and Appendix G.
> Because of the additional experiments that have been requested, we are still working on the clean version of the code, and we will make sure to provide the code repository in the camera-ready version once it is published, and we are convinced that it is the common practice agreed by the community.

---

### Author Response · Authors · 2024-11-24
**General response && We would like to hear your feedback**

Dear reviewers,

We would like to thank you for taking time to review our paper and provide extremely valuable and constructive feedback. The discussions raised by each of you have been very fruitful for us to improve both presentation and experimental completeness of our paper. We would like to further answer some common questions and highlight the major changes in our revised manuscript. We are convinced that we have addressed all your concerns and would be delighted to hear your thoughts on our response and the revised manuscript.

1. $\textbf{Novelty/Contribution:}$ GIVE is a novel reasoning framework inspired from human cognitive process, which is to use the limited expert information as high-level hints for LLM to conduct divergent thinking, thus solve the reasoning-intensive query. We are glad that all reviewers recognize the motivation of our paper: We provide an approach (GIVE) for LLM to retrieve and reason on sparse KGs, whereas all previous methods fail heavily because (1) LLM does not have enough self-knowledge to solve the query (CoT). (2) There is no direct information in the external knowledge source to directly solve the query (ToG).

2. $\textbf{Inclusiveness of experiments:}$ We include KGs of different sizes and sparsities: small and dense(umls), large and sparse(10% / 50% ConceptNet), large and dense (full ConceptNet), QA datasets from different domains (biomedical, commonsense, open-domain), of different types (yes-no, multiple-choice, text generation), LLMs of different sizes (LLama3.1, GPT3.5T, GPT4, GPT4o-mini). In every one of these comparisons, GIVE achieves the best performance. For every experimental setting, we clarify in both the comments and the revised manuscript the reason to choose this setting: We aim to test GIVE’s ability in boosting the model’s reasoning ability, especially when it is hard to get enough knowledge during training.

3. $\textbf{Efficiency of GIVE:}$ We include detailed analyze and numerical results on GIVE’s efficiency in Appendix F.2 of the revised manuscript, covering every factor that could potentially influence GIVE’s efficiency in both time and context length. Reasoning guarantees better performance, but at the same time, takes time (like human).  GIVE (and its efficient alternative) achieves balanced tradeoff between efficiency and accuracy in these hard tasks. The results itself proved that the knowledge generated by GIVE is of high quality.

$\textbf{Summery of Changes:}$

  $\textbf{Experiments:}$ $\textbf{New baseline}$: GraphRAG (Section 4.2); $\textbf{New dataset}$: TruthfulQA (Appendix D); $\textbf{New ablation studies}$: inner/inter group connections (Appendix E.1); $\textbf{Additional detailed efficiency comparison}$: time/context length (Appendix F.2)

$\textbf{Presentation:}$ formalize description of Section 3.3 and 3.5, clarification of "sparse" in Section 4.1.1 and change description of UMLS from “sparse” to be “small and dense”, quantify all datasets in Appendix C, discussion about connections/difference between GIVE and RAG in Appendix B, more clear motivation statement in abstract and conclusion, as well as all the other new findings from the discussions with the reviewers.

---

### Public Comment · ~Surya_Tej_Kodali1 · 2024-12-02

I would like to share how the GIVE framework has significantly improved our company's medical coding and billing processes. In our industry, we often deal with incomplete or proprietary insurance guidelines. Companies like InterQual and MCG do not publicly disclose their guidelines, resulting in sparse and sometimes inconsistent information. While these guidelines may broadly align with Medicare policies, critical differences exist that we must accurately identify and apply.

By implementing GIVE, we developed a customized solution that integrates sparse knowledge graphs with our existing systems. This framework allowed us to:

- **Model Complex Relationships:** GIVE helped us understand intricate dependencies between medical codes, such as identifying mutually exclusive codes and handling conditional billing rules. This is crucial when certain procedures cannot be billed together or when specific guidelines take precedence.

- **Handle Data Sparsity Effectively:** Even with limited external information, GIVE enabled us to combine our internal knowledge with sparse external data, filling gaps in proprietary or incomplete guidelines. This was instrumental in adapting to various insurance-specific rules that deviate from standard Medicare guidelines.

- **Improve Efficiency Through Pruning:** To address the challenge of numerous 2-hop neighbors in the knowledge graph, we utilized additional metadata and embeddings to prune irrelevant results. This optimization allowed us to focus on the most pertinent relationships without compromising efficiency, ensuring the framework scaled well with larger datasets.

- **Enhance Compliance and Accuracy:** By accurately modeling and prioritizing guidelines from CMS, NCCI, and individual insurance providers, GIVE reduced the risk of claim denials and audits. It ensured that we consistently applied the most relevant policies in our coding and billing practices.

Implementing GIVE has led to improved coding accuracy, enhanced compliance with regulatory requirements, and increased operational efficiency. The framework's ability to handle data sparsity and complex relationships has been invaluable in our real-world setting, where guidelines are often incomplete or proprietary.

Our successful application of GIVE underscores its practical applicability and effectiveness in specialized domains.

---

> ### Author Response · Authors · 2024-12-02
> **Thank you for your comments!**
>
> Hi Surya,
>
> Thank you for your comments! Your feedback means a lot.
>
> We are glad that you find GIVE useful in your evaluations and we feel blessed that our research helped in the real-world applications to boost LLM's performance with sparse external knowledge! This is exactly the research question we would like to answer in this paper.

---

### Meta-Review · Area_Chair_zvrn · 2024-12-18

**Metareview:**

This paper introduces GIVE, a framework leveraging sparse knowledge graphs (KGs) to enhance large language models (LLMs) for reasoning tasks. While the reviewers acknowledge that the approach is well motivated, several critical shortcomings are identified:

- Limited evaluation scope: The experiments focus on specialized tasks, like biomedical QA, with insufficient general-domain benchmarks (qXMg, Z1Eq).
- Scalability concerns: The computational complexity for larger and denser KGs remains unaddressed, raising concerns about practical applicability (qXMg, Z1Eq).
- Baseline issues: Comparisons with relevant methods, such as GraphRAG, are incomplete, and the choice of baselines leaves gaps in experimental rigor (7idj, qXMg).
- Conflict resolution: Mechanisms for handling discrepancies between KG-derived and LLM-internal knowledge are inadequately explored (Z1Eq).

**Additional Comments On Reviewer Discussion:**

The authors addressed some issues during the rebuttal, including additional experiments and clarifications. However, after the author-reviewer discussion and reviewer-AC discussions, the reviewers are still concerned about the limited transferability, relatively narrow scope of the paper, dataset and setting choices in the experiments (qXMg, Z1Eq).

Due to its limited scope, unresolved scalability issues, and incomplete comparisons, a rejection is recommended. The authors are encouraged to expand the evaluation and refine scalability in future work.

---

### Decision · Program_Chairs · 2025-01-22

Reject